# **bears** Make Neuro-Symbolic Models Aware of their Reasoning Shortcuts

Emanuele Marconato[1,2,*]    Samuele Bortolotti[1,*]    Emile van Krieken[3,*]    Antonio Vergari[3]    Andrea Passerini[1]

Stefano Teso[1,4]

[1]Department of Information Engineering and Computer Science , University of Trento , Trento, Italy
[2]Department of Computer Science, University of Pisa, Pisa, Italy
[3]School of Informatics, University of Edinburgh, Edinburgh, United Kingdom
[4]CIMeC, University of Trento, Rovereto, Italy
* Equal contribution.

## Abstract

Neuro-Symbolic (NeSy) predictors that conform to symbolic knowledge – encoding, *e.g.*, safety constraints – can be affected by Reasoning Shortcuts (RSs): They learn concepts consistent with the symbolic knowledge by exploiting unintended semantics. RSs compromise reliability and generalization and, as we show in this paper, they are linked to NeSy models being overconfident about the predicted concepts. Unfortunately, the only trustworthy mitigation strategy requires collecting costly dense supervision over the concepts. Rather than attempting to avoid RSs altogether, we propose to ensure NeSy models are *aware of the semantic ambiguity of the concepts they learn*, thus enabling their users to identify and distrust low-quality concepts. Starting from three simple desiderata, we derive bears (BE Aware of Reasoning Shortcuts), an ensembling technique that calibrates the model's concept-level confidence without compromising prediction accuracy, thus encouraging NeSy architectures to be uncertain about concepts affected by RSs. We show empirically that bears improves RS-awareness of several state-of-the-art NeSy models, and also facilitates acquiring informative dense annotations for mitigation purposes.

## 1 INTRODUCTION

Research in Neuro-Symbolic (NeSy) AI [4, 5, 6] has recently yielded a wealth of architectures capable of integrating low-level perception and symbolic reasoning. Crucially, these architectures encourage [7] or guarantee [8, 9, 10, 11] that their predictions conform to given prior knowledge encoding, *e.g.*, structural or safety constraints, thus offering improved reliability compared to neural baselines [12, 11, 9].

It was recently shown that, however, NeSy architectures can achieve high prediction accuracy by learning concepts – *aka* neural predicates [8] – with unintended semantics [3, 13]. E.g., consider an autonomous driving task like BDD-OIA [1] in which a model has to predict safe actions based on the contents of a dashcam image, under the constraint that whenever it detects pedestrian or red lights the vehicle must stop. Then, the model can achieve perfect accuracy *and* comply with the constraint even when confusing pedestrians for red lights, precisely because both entail the correct (stop) action [3]. See Fig. 1 for an illustration.

These so-called *reasoning shortcuts* (RSs) occur because the prior knowledge and data may be insufficient to pin-down the intended semantics of all concepts, and cannot be avoided by maximizing prediction accuracy alone. They compromise in-distribution [14] and out-of-distribution generalization [3, 13], continual learning [15], reliability of neuro-symbolic verification tools [16], and concept-based interpretability [17, 18] and debugging [19]. Importantly, unsupervised mitigation strategies either offer no guarantees or work under restrictive assumptions, while supervised ones involve acquiring costly side information, *e.g.*, concept supervision [3].

Rather than attempting to avoid RSs altogether, we suggest NeSy predictors should be *aware of their reasoning shortcuts*, that is, they should assign lower confidence to concepts affected by RSs, thus enabling users to identify and avoid low-quality predictions, all while retaining high accuracy. Unfortunately, as we show empirically, state-of-the-art NeSy architectures are *not* RS-aware. We address this issue by introducing bears (BE Aware of Reasoning Shortcuts), a simple but effective method for making NeSy predictors RS-aware that does *not* rely on costly dense supervision. bears replaces the concept extraction module with a diversified *ensemble* specifically trained to encourage the concepts' uncertainty is proportional to how strongly these are impacted by RSs. Our experiments show that bears successfully improves RS-awareness of three state-of-the-art NeSy architectures on four NeSy data sets, including

*Accepted for the 40th Conference on Uncertainty in Artificial Intelligence* (UAI 2024).

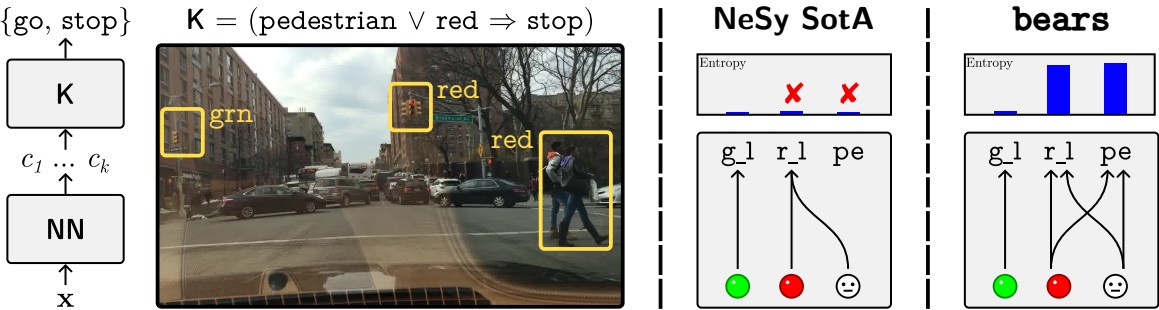

Figure 1: **bears lessens overconfidence due to reasoning shortcuts. Left**: In the BDD-OIA autonomous driving task [1, 2], NeSy predictors can attain high accuracy *and* comply with the knowledge even when confusing the concepts of pedestrian (ped) and red light (red) [3]. **Middle**: State-of-the-art NeSy architectures predict concepts affected by RSs with high confidence, making it impossible to discriminate between reliable and unreliable concept predictions. **Right**: bears encourages them to allocate probability to *conflicting* concept maps, substantially lessening overconfidence.

a high-stakes autonomous driving task, and enables us to design a simple but effective active learning strategy for acquiring concept annotations for mitigation purposes.

**Contributions**. Summarizing, we:

- Shift focus from RS mitigation to RS awareness and show that state-of-the-art NeSy predictors are not RS-aware.
- Propose bears, which improves RS-awareness of NeSy predictors without relying on dense supervision.
- Demonstrate that it outperforms SotA uncertainty calibration methods on several tasks and architectures.
- Show that it enables intelligent acquisition of concept annotations, thus lowering the cost of supervised mitigation.

## 2 PRELIMINARIES

**Notation.** We denote scalar constants $x$ in lower-case, random variables $X$ in upper case, and ordered sets of constants $\mathbf{x}$ and random variables $\mathbf{X}$ in bold typeface. Throughout, we use the shorthand $[n] := \{1, \ldots, n\}$.

**Neuro-Symbolic Predictors.** RSs have been primarily studied in the context of *NeSy predictors* [20, 21], which we briefly overview next. Given an input $\mathbf{x} \in \mathbb{R}^n$, these models infer a (multi-)label $\mathbf{y} \in \{0,1\}^m$ by leveraging *prior knowledge* K encoding, *e.g.*, known structural [12] or safety [11] constraints. During inference, they first extract a set of *concepts* $\mathbf{c} \in \{0,1\}^k$ using a (neural) concept extractor $p_\theta(\mathbf{C} \mid \mathbf{x})$. Then, they reason over these to obtain a predictive distribution $p_\theta(\mathbf{y} \mid \mathbf{c}; \mathsf{K})$ that associates lower [7, 22, 23] or provably zero [8, 10] probability to outputs $\mathbf{y}$ that violate the knowledge K. Taken together, these two distributions define a NeSy predictor of the form $p_\theta(\mathbf{y} \mid \mathbf{x}; \mathsf{K})$. The complete pipeline is visualized in Fig. 1 (left).

**Example 1.** *In our running example (Fig. 1), given a dash-cam image $\mathbf{x}$, we wish to infer what action $y \in \{\text{stop}, \text{go}\}$ a vehicle should perform. This task can be modelled using* three binary concepts $C_1, C_2, C_3$ *encoding the presence of green lights (*grn*), red lights (*red*), and pedestrians (*ped*). The knowledge specifies that if any of the latter two is detected, the vehicle must stop:* $\mathsf{K} = (\text{ped} \lor \text{red} \Rightarrow \text{stop})$.

Inference amounts to solving a MAP [24] problem $\operatorname{argmax}_\mathbf{y} p_\theta(\mathbf{y} \mid \mathbf{x}; \mathsf{K})$, and learning to maximize the log-likelihood on a training set $\mathcal{D} = \{(\mathbf{x}_i, \mathbf{y}_i)\}$. Architectures chiefly differ in how they integrate the concept extractor and the reasoning layer, and in whether inference and learning are exact or approximate, see Section 5 for an overview. Despite these differences, RSs are a general phenomenon that can affect all NeSy predictors [3].

**Reasoning Shortcuts.** In NeSy, usually only the labels receive supervision, while the concepts are treated as *latent variables*. It was recently shown that, as a result, NeSy models can fall prey to *reasoning shortcuts* (RSs), *i.e.*, they often achieve high label accuracy by learning concepts with unintended semantics [3, 15, 14, 13].

To properly understand RSs, we need to define how the data is generated, cf. Fig. 2. Following [3], we assume there exist $k$ unobserved *ground-truth concepts* $\mathbf{g} \in \{0,1\}^k$ drawn from a distribution $p^*(\mathbf{G})$, which generate both the observed inputs $\mathbf{x}$ and the label $\mathbf{y}$ according to unobserved distributions $p^*(\mathbf{X} \mid \mathbf{G})$ and $p^*(\mathbf{Y} \mid \mathbf{G}; \mathsf{K})$, respectively. We also assume all observed labels satisfy the prior knowledge K given $\mathbf{g}$.

In essence, a NeSy predictor is affected by an RS whenever the label distribution $p_\theta(\mathbf{Y} \mid \mathbf{X}; \mathsf{K})$ behaves well, but the concept distribution $p_\theta(\mathbf{C} \mid \mathbf{X})$ does not, that is, given inputs $\mathbf{x}$ it extracts concepts $\mathbf{c}$ that yield the correct label $\mathbf{y}$ but do not match the ground-truth ones $\mathbf{g}$. RSs impact the reliability of learned concepts and thus the trustworthiness of NeSy architectures in out-of-distribution [3] scenarios, continual learning [15] settings, and neuro-symbolic verification [16]. They also compromise the interpretability of

concept-based explanations of the model's inference process [25, 26, 18].

**Example 2.** *In Example 1, we would expect predictors achieving high label accuracy to accurately classify all concepts, too. It turns out that, however, predictors misclassifying pedestrians as red lights (as in Fig. 1, middle) can achieve an equally high label accuracy, precisely because both concepts entail the (correct)* `stop` *action according to* K. *To see why this is problematic, consider tasks where the knowledge allows for ignoring red lights when there is an emergency. This can lead to dangerous decisions when there are pedestrians on the road [3].*

**Causes and Mitigation Strategies.** The factors controlling the occurrence of RSs include [3]: 1) The structure of the *knowledge* K, 2) The distribution of the *training data*, 3) The learning *objective*, and 4) The *architecture* of the concept extractor. For instance, whenever the knowledge K admits multiple solutions – that is, the correct label $\mathbf{y}$ can be inferred from distinct concept vectors $\mathbf{c} \neq \mathbf{c}'$, as in Example 2 – the NeSy model has no incentive to prefer one over the other, as they achieve exactly the same likelihood on the training data, and therefore may end up learning concepts that do not match the ground-truth ones [15].

All four root causes are natural targets for mitigation. For instance, one can reduce the set of unintended solutions admitted by the knowledge via *multi-task learning* [27], force the model to distinguish between different concepts by introducing a *reconstruction penalty* [28], and reduce ambiguity by ensuring the concept encoder is *disentangled* [29]. It was shown theoretically and empirically that, while existing unsupervised mitigation strategies *can* and in fact *do* have an impact on the number of RSs affecting NeSy predictors, especially when used in combination, they are also insufficient to prevent RSs in all applications [3].

The most direct mitigation strategy is that of supplying *dense annotations* for the concepts themselves. Doing so steers the model towards acquiring good concepts and can, in fact, prevent RSs [3], yet concept supervision is expensive to acquire and therefore rarely available in applications.

## 3 FROM MITIGATION TO AWARENESS

Mitigating RSs is highly non-trivial. Rather than facing this issue head on, we propose to make NeSy predictors *reasoning shortcut-aware*, *i.e.*, uncertain about concepts with ambiguous or wrong semantics. To see why this is beneficial, consider the following example:

**Example 3.** *Imagine a NeSy predictor that confuses pedestrians with red lights, as in Example 2. If it is always certain about its concept-level predictions, as in Fig. 1 (middle), there is no way for users to figure out that some predictions should not be trusted. A model classifying pedestrians as*

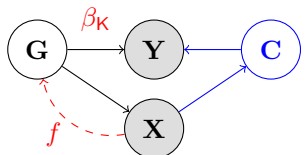

Figure 2: **Data generating process**. The (unobserved) ground-truth concepts $\mathbf{G}$ cause the inputs $\mathbf{X}$ which cause the labels $\mathbf{Y}$ (in **black**). A NeSy predictor learns to map inputs $\mathbf{X}$ to concepts $\mathbf{C}$ (in **blue**), which ideally should match the concepts $\mathbf{G}$ that caused $\mathbf{X}$. The maps $f$ and $\beta_\mathsf{K}$ from assumptions **A1** and **A2** in Section 3 are shown in **red**.

*both* `ped`*estrians and* `red` *lights with equal probability, as in Fig. 1 (right), is just as confused, but is also calibrated, in that it is more uncertain about pedestrians and red lights, which are low quality, compared to green lights, which are classified correctly. This enables users to distinguish between high- and low-quality predictions and concepts, and thus avoid the latter.*

We say a NeSy predictor is *reasoning shortcut-aware* if it satisfies the following desiderata:

**D1**. **Calibration**: For all concepts *not* affected by RSs, the system should achieve high accuracy and be highly confident. Vice versa, for all concepts that *are* affected by RSs, the model should have low confidence.

**D2**. **Performance**: The predictor $p_\theta(\mathbf{Y} \mid \mathbf{X}; \mathsf{K})$ should achieve high *label accuracy* even if RSs are present.

**D3**. **Cost effectiveness**: The system should not rely on expensive mitigation strategies.

Calibration (**D1**) captures the essence of our proposal: a model that knows which ones of the learned concepts are affected by RSs can prevent its users from blindly trusting and reusing them. Naturally, this should not come at the cost of prediction accuracy (**D2**) or expensive concept-level annotations (**D3**), so as not to hinder applicability.

### 3.1 AWARENESS VIA ENTROPY MAXIMIZATION

We start by introducing our basic intuition. Consider an example $(\mathbf{x}, \mathbf{y})$ and let $\mathbf{g}$ be the underlying ground-truth concepts. If $\mathbf{g}$ is the only concept vector that entails the label $\mathbf{y}$ according to the prior knowledge K, maximizing the likelihood steers the concept extractor towards predicting the correct concept $\mathbf{c} = \mathbf{g}$ with high confidence. In this simplified scenario, NeSy predictors would automatically satisfy **D1**–**D3**. In most NeSy tasks, however, there exist multiple concept vectors $\mathbf{c}_1 \neq \ldots \neq \mathbf{c}_u$ that *all* entail the correct label $\mathbf{y}$. In this case, there is no reason for the model to prefer one to the others: all of them achieve the same (optimal) likelihood, yet only one of them matches $\mathbf{g}$. The

issue at hand is that existing NeSy predictors tend to predict only one – likely incorrect – $\mathbf{c}_i$, $i \in [u]$, and they do so *with high confidence*, thus falling short of **D1**.

We propose an alternative solution. Let $\Theta^*$ be the set of parameters $\theta$ attaining high accuracy (**D2**), *i.e.*, mapping inputs to concepts $\mathbf{c}$ yielding good predictions $\mathbf{y}$. This set includes the (correct) predictor mapping $\mathbf{x}$ into $\mathbf{g}$ as well as all high-performance predictors mapping it to one or more unintended concept vectors $\mathbf{c}_i \neq \mathbf{g}$. We wish to find one that is maximally uncertain about which $\mathbf{c}_i$'s it should output, that is:

$$\max_{\theta \in \Theta^*} H(p_\theta(\mathbf{C} \mid \mathbf{G})) \tag{1}$$

Here, $H(p_\theta(\mathbf{C} \mid \mathbf{G})) = -\mathbb{E}_{p^*(\mathbf{g}) p_\theta(\mathbf{c}|\mathbf{g})}[\log p_\theta(\mathbf{c} \mid \mathbf{g})]$ is the conditional Shannon entropy, and:

$$p_\theta(\mathbf{C} \mid \mathbf{G}) := \int p_\theta(\mathbf{C} \mid \mathbf{x}) \, p^*(\mathbf{x} \mid \mathbf{G}) \, \mathrm{d}\mathbf{x} \tag{2}$$

is the distribution obtained by marginalizing the concept extractor $p_\theta(\mathbf{C} \mid \mathbf{X})$ over the inputs $\mathbf{x}$. By construction, this $\theta$ achieves high accuracy (**D2**) but, despite being affected by RSs, it is less confident about its concepts $\mathbf{c}$ (**D1**). The issue is that, by **D3**, we have access to neither the ground-truth distribution $p_\theta(\mathbf{C} \mid \mathbf{G})$ nor to samples drawn from it, so we cannot optimize Eq. (1) directly.

## 3.2 MAXIMIZING ENTROPY

Next, we show that one can maximize Eq. (1) by constructing a distribution $p_\theta(\mathbf{C} \mid \mathbf{G})$ affected by *multiple* but *conflicting* RSs. Our analysis builds on that of [3], which relies on two simplifying assumptions:

**A1**. *Invertibility*: Each $\mathbf{x}$ is generated by a unique $\mathbf{g}$, *i.e.*, there exists a function $f : \mathbf{x} \mapsto \mathbf{g}$ such that $p^*(\mathbf{G} \mid \mathbf{X}) = \mathbb{1}\{\mathbf{G} - f(\mathbf{X})\}$.[1]

**A2**. *Determinism*: The knowledge K is *deterministic*, *i.e.*, there exists a function $\beta_\mathsf{K} : \mathbf{g} \mapsto \mathbf{y}$ such that $p^*(\mathbf{Y} \mid \mathbf{G}; \mathsf{K}) = \mathbb{1}\{\mathbf{Y} = \beta_\mathsf{K}(\mathbf{g})\}$. This is often the case in NeSy tasks, *e.g.*, BDD-OIA .

The link between $\beta_\mathsf{K}$, $f$, and the NeSy predictor is shown in Fig. 2. In the following, we use $\alpha : \{0,1\}^k \to \{0,1\}^k$ to indicate a generic map from $\mathbf{g}$ to $\mathbf{c}$, and denote $\mathcal{A}$ the set of all such maps and $\mathcal{A}^* \subseteq \mathcal{A}$ that of $\alpha$'s that yield distributions $p_\theta(\mathbf{C} \mid \mathbf{G})$ achieving perfect label accuracy (**D2**). Each $\alpha$ encodes a corresponding *deterministic* distribution $p_\theta(\mathbf{C} \mid \mathbf{G}) = \mathbb{1}\{\mathbf{c} = \alpha(\mathbf{g})\}$: if $\alpha$ is the identity, this distribution encodes the correct semantics. Otherwise, it captures an RS (cf. Fig. 1). Next, we show that every distribution $p_\theta(\mathbf{C} \mid \mathbf{G})$ decomposes as a convex combination of maps $\alpha$.

---

[1]Works on the *identifiability* of the latent variables in independent component analysis [28, 30, 31] and causal representation learning [29, 32, 33] build on a similar assumption.

**Lemma 1.** *For any $p(\mathbf{C} \mid \mathbf{G})$, there exists at least one vector $\boldsymbol{\omega}$ such that the following holds:*

$$p(\mathbf{C} \mid \mathbf{G}) = \sum_{\alpha \in \mathcal{A}} \omega_\alpha \mathbb{1}\{\mathbf{C} = \alpha(\mathbf{G})\} := p_\omega(\mathbf{C} \mid \mathbf{G}) \tag{3}$$

*where $\boldsymbol{\omega} \geq 0$, $\|\boldsymbol{\omega}\|_1 = 1$. Crucially, under invertibility (**A1**) and determinism (**A2**), if $p_\theta(\mathbf{C} \mid \mathbf{G})$ is optimal (**D2**), Eq. (3) holds even if we replace $\mathcal{A}$ with $\mathcal{A}^*$.*

All proofs can be found in Appendix B. This means that most distributions $p(\mathbf{C} \mid \mathbf{G})$ are mixtures of *multiple* maps $\alpha \in \mathcal{A}^*$, each potentially capturing a different RS. Naturally, if $\omega_\alpha$ is non-zero only for those $\alpha$'s that fall in $\mathcal{A}^*$, $p_\theta(\mathbf{C} \mid \mathbf{G})$ achieves high performance (**D2**). The question is what $\boldsymbol{\omega}$'s achieve calibration (**D1**). Intuitively, if $\boldsymbol{\omega}$ mixes $\alpha$'s capturing RSs that disagree on the semantics of some concepts and agree on others, $p_\theta(\mathbf{C} \mid \mathbf{G})$ is RS-aware.

**Example 4.** *Consider Fig. 1 and two high-performance maps in $\mathcal{A}^*$: $\alpha_1$ mapping green lights to* grn*, and both pedestrians and red lights to* red*; $\alpha_2$ also mapping green lights to* grn*, but pedestrians and red lights to* ped*. Clearly, both maps are affected by RSs and overconfident, yet their mixture $\alpha = \frac{1}{2}(\alpha_1 + \alpha_2)$ yields a distribution $p_\theta(\mathbf{C} \mid \mathbf{G})$ that looks exactly like the one in Fig. 1 (right), which predicts* grn *correctly with high confidence, and* red *and* ped *with low confidence, and thus satisfies **D1** and **D2**.*

In other words, this allows us to leverage the model's uncertainty to estimate the extent by which concepts are affected by RSs *without the need for dense annotations* (**D3**). Due to space considerations, we report our formal analysis of the connection between uncertainty and RSs in Appendix B.2. The next result indicates that this intuition is consistent with our original objective in Eq. (1):

**Proposition 2.** *(Informal.) If $p_\theta$ is expressive enough, under invertibility (**A1**) and determinism (**A2**), it holds that:*

$$\max_{\theta \in \Theta^*} H(p_\theta(\mathbf{C} \mid \mathbf{G})) = \max_{\boldsymbol{\omega}^*} H(p_{\boldsymbol{\omega}^*}(\mathbf{C} \mid \mathbf{G})) \tag{4}$$

This also tells us that we can solve Eq. (1) by finding a combination of maps $\alpha$'s with maximal entropy over concepts.

## 3.3 RS-AWARENESS WITH BEARS

Our results suggest that RS-awareness can be achieved by constructing an ensemble $\boldsymbol{\theta} = \{\theta_i\}$ of (deterministic) high-performance concept extractors affected by distinct RSs. Ideally, we could construct such an ensemble by training multiple concept extractors $p_{\theta_i}(\mathbf{C} \mid \mathbf{X})$ such that each of them picks up a different RS, and then defining an overall predictor $p_\theta$ as a convex combination thereof, that is, $p_\theta(\mathbf{C} \mid \mathbf{X}) = \sum_i \lambda_i p_{\theta_i}(\mathbf{C} \mid \mathbf{X})$, where $\boldsymbol{\lambda} \geq 0$ and $\|\boldsymbol{\lambda}\|_1 = 1$. We next show that, if the ensemble is large enough, such a model does optimize our original objective in Eq. (1).

**Proposition 3.** *(Informal.) Let $p(\mathbf{C} \mid \mathbf{X})$ be a convex combination of models $p_{\theta_i}(\mathbf{C} \mid \mathbf{X})$ with parameters $\boldsymbol{\theta} = \{\theta_i\}$ and weights $\boldsymbol{\lambda} = \{\lambda_i\}$, such that $\theta_i \in \Theta^*$. Under invertibility and determinism, there exists a $K \leq |\mathcal{A}^*|$ such that for an ensemble with $K$ members, it holds that:*

$$\max_{\boldsymbol{\theta}, \boldsymbol{\lambda}} H\Big( \sum_{i=1}^{K} \lambda_i p_{\theta_i}(\mathbf{C} \mid \mathbf{X}) \Big) = \max_{\boldsymbol{\omega}^*} H(p_{\boldsymbol{\omega}^*}(\mathbf{C} \mid \mathbf{G})) \quad (5)$$

*Moreover, maximizing $H(p_{\boldsymbol{\theta}}(\mathbf{C} \mid \mathbf{X}))$ amounts to solving:*

$$\max_{\boldsymbol{\theta}, \boldsymbol{\lambda}} \int p(\mathbf{x}) \sum_{i=1}^{K} \lambda_i [\mathsf{KL}(p_{\theta_i}(\mathbf{c} \mid \mathbf{x}) \mid\mid \sum_{j=1}^{K} \lambda_j p_{\theta_j}(\mathbf{c} \mid \mathbf{x}))$$
$$+ H(p_{\theta_i}(\mathbf{C} \mid \mathbf{x}))]\mathrm{d}\mathbf{x} \quad (6)$$

*where* $\mathsf{KL}$ *denotes the Kullback-Lieber divergence.*

For the proposition to apply, it may be necessary to collect an enormous number of diverse, deterministic, high-performance models, potentially as many as $|\mathcal{A}^*|$. Naturally, constructing such an ensemble is highly impractical. Thankfully, doing so is often unnecessary in practice: as long as the ensemble contains models that disagree on the semantics of concepts, it will likely achieve high entropy on concepts affected by RSs and low entropy on the rest, as we show in our experiments.

bears exploits this observation to turn this into a practical algorithm. In short, it grows an ensemble $\boldsymbol{\theta}$ by optimizing a joint training objective combining label accuracy and diversity of concept distributions. Each model $\theta_i$ is learned in turn by maximizing the following quantity:

$$\mathcal{L}(\mathbf{x}, \mathbf{y}; \mathsf{K}, \theta_t) + \gamma_1 \cdot \mathsf{KL}\big(p_{\theta_t}(\mathbf{C} \mid \mathbf{x}) \mid\mid \frac{1}{t} \sum_{j=1}^{t} p_{\theta_j}(\mathbf{C} \mid \mathbf{x})\big)$$
$$+ \gamma_2 \cdot H(p_{\theta_t}(\mathbf{C} \mid \mathbf{x})) \quad (7)$$

over a training set $\mathcal{D}$. Here, $\mathcal{L}(\mathbf{x}, \mathbf{y}; \mathsf{K}, \theta)$ is the log-likelihood of member $\theta_i$, while the second term is a $\mathsf{KL}$ divergence – obtained from Eq. (6) by taking a uniform $\boldsymbol{\lambda}$ – encouraging $\theta_i$ to differ from $\theta_1, \ldots, \theta_{i-1}$ in terms of concept distribution. Finally, $\gamma_1$ and $\gamma_2$ are hyperparameters. Pseudocode and further details on the $\mathsf{KL}$ term can be found in Appendix A. We remark that, despite learning $\theta_i$'s that are not necessarily optimal or deterministic, in practice bears still manages to drastically improve RS-awareness in our experiments.

### 3.4 BEARS THROUGH A BAYESIAN LENS

Bayesian inference is a popular strategy for lessening over-confidence of neural networks [34, 35, 36]. It works by marginalizing over a (possibly uncountable) family of alternative predictors, each weighted according to a posterior

distribution $p(\theta \mid \mathcal{D}) \propto p(\mathcal{D} \mid \theta) \cdot p(\theta)$ accounting for both data fit $p(\mathcal{D} \mid \theta)$ and prior information $p(\theta)$. Formally, the label distribution is given by:

$$p(\mathbf{y} \mid \mathbf{x}; \mathcal{D}) = \int p_\theta(\mathbf{y} \mid \mathbf{x}; \mathsf{K}) \cdot p(\theta \mid \mathcal{D}) \, \mathrm{d}\theta \quad (8)$$

The expectation is computationally intractable and thus often approximated in practice. E.g., Monte Carlo approaches compute an unbiased estimate of Eq. (8) by averaging the label distribution $p_{\theta_i}(\mathbf{y} \mid \mathbf{x}; \mathsf{K})$ of a (small) selection of parameters $\{\theta_i\}$. More advanced Bayesian techniques for neural networks [36], like the Laplace approximation [37] and variational inference methods [38], locally approximate the posterior around the (parameters of the) trained model. Conceptually simpler techniques like deep ensembles [39] average over a bag of diverse neural networks trained in parallel and have proven to be surprisingly effective in practice.

Recall that, by Eq. (7), bears averages over models $\theta_i \in \Theta^*$ that achieve high likelihood but disagree in terms of concepts. This can be viewed as a form of Bayesian inference. Specifically, Eq. (8) behaves similarly to bears if we select the prior and likelihood appropriately. In fact, if 1) the prior $p(\theta)$ associates non-zero probability to all $\theta$'s encoding an RS, and 2) the likelihood $p(\mathcal{D} \mid \theta)$ allocates non-zero probability only to $\theta$'s that match the data (almost) perfectly, the resulting posterior $p(\theta \mid \mathcal{D})$ associates probability mass only to models that satisfy **D1** and **D2**, that is, those in $\Theta^*$.

Compared to stock Bayesian techniques, bears is specifically designed to handle RSs. First, note that the likelihood $p(\mathcal{D} \mid \theta)$ is highly multimodal, as it peaks on the "optimal" models in $\Theta^*$, thus Bayesian techniques that focus on neighborhood of trained networks have trouble recovering all modes [40]. Moreover, the expectation in Eq. (8) runs over parameters $\theta$, which may be redundant, in the sense that different $\theta_i$'s can entail similar or identical concept encoders $p_\theta(\mathbf{C} \mid \mathbf{X})$. This suggests that covering the space of $\theta$'s, as done in [41, 42], is sub-optimal compared to averaging over $\theta_i$'s that disagree on which concepts they predict. bears is designed to avoid both issues, as it learns models $\{\theta_i\}$ that have both high likelihood and disagree on the semantics of concepts, so as to capture multiple, different modes of the likelihood, thus encouraging RS-awareness.

### 3.5 ACTIVE LEARNING WITH DENSE ANNOTATIONS

As mentioned in Section 2, a sure-proof way of avoiding RSs is to leverage concept-level annotations, which are however expensive to acquire. bears helps to address this issue. Specifically, we propose to exploit the model's concept-level uncertainty – which is higher for the concepts most affected by RSs – to implement a cost-efficient annotation acquisition strategy.

We consider the following scenario: given a NeSy predictor $p_\theta$ affected by RSs and a pool of examples $\mathcal{D} = \{(\mathbf{x}_i, \mathbf{y}_i)\}$, we seek to mitigate RSs by eliciting concept-level annotations for as few data points as possible. This immediately suggests leveraging active learning techniques to select informative data points [43]. Options include selecting examples $(\mathbf{x}_i, \mathbf{y}_i)$ in $\mathcal{D}$ with the highest concept entropy $H(p(\mathbf{C} \mid \mathbf{x}_i))$ and requesting dense annotations for the entire concept vector $\mathbf{G}_i$, or requesting supervision only for specific concepts $G_j$ by maximizing $H(p(C_j \mid \mathbf{x}_i))$ for $i$ and $j$. Both entropies are cheap to compute for most neural networks (as the predicted concepts $C_j$ are conditionally independent given the input $\mathbf{x}$ [44]), making acquisition both practical and easy to set up.

Crucially, these strategies only work if the model is RS-aware and, in fact, they fail for state-of-the-art NeSy architectures unless paired with bears, as shown in Section 4.

### 3.6 BENEFITS AND LIMITATIONS

The most immediate benefit of bears – and of RS-awareness in general – is that it enables users to identify and avoid untrustworthy predictions $\mathbf{c}$ or even individual concepts $c_i$, substantially improving the reliability of NeSy pipelines. Moreover, compared to simpler Bayesian approaches for uncertainty calibration, it is specifically designed for dealing with the multimodal nature of the RS landscape and – as shown by our experiments – yields more calibrated concept uncertainty in practice. Finally, bears enables leveraging the model's uncertainty estimates to guide elicitation of concept supervision.

A downside of bears is that training time grows (linearly) with the size of the ensemble $\boldsymbol{\theta}$. This extra cost is justified in tasks where reliability matters, such as high-stakes applications or when learning concepts for model verification [16]. Regardless, in our experiments, an ensemble of 5-10 concept extractors is sufficient to dramatically improve RS-awareness compared to regular NeSy predictors, with a runtime cost comparable to alternative calibration approaches. This is not too surprising: in principle, even an ensemble of *two* models is sufficient to ensure improved calibration, provided these capture RSs holding strong contrasting beliefs. Finally, bears involves two other hyperparameters: $\gamma_1$ and $\gamma_2$, which can be tuned, *e.g.*, via cross-validation on a validation split. As for the relative importance of different members of the ensemble (that is, $\boldsymbol{\lambda}$), our experiments suggest that even taking a uniform average already substantially improves RS-awareness compared to existing approaches.

## 4 EMPIRICAL ANALYSIS

In this section, we tackle the following research questions:

**Q1**. Are existing NeSy predictors RS-aware?

**Q2**. Does bears make NeSy predictors RS-aware?

**Q3**. Does bears facilitate acquiring informative concept-level supervision?

To answer them, we evaluate three state-of-the-art NeSy architectures before and after applying bears and other well-known uncertainty calibration methods. Our code can be found at: https://github.com/samuelebortolotti/bears.

**NeSy predictors.** We consider the following architectures.

DeepProbLog (DPL) [8] instantiates one neural predicate for each of the binary concepts present in the knowledge, and implements them using one or more neural networks feeding on the input. It then predicts a label by combining the neural predicates via probabilistic logic reasoning [45]. It speeds up this step by compiling inference into a computational circuit using knowledge compilation techniques [44]. This circuit contains no trainable parameter in our experiments.

Semantic Loss (SL) [7] is a penalty term used to encourage deep neural entworks to output predictions consistent with given prior knowledge. In our experiments, we use a setup similar to similar to Concept Bottleneck Models [46]: we employ a neural network to predict the concepts, whose logits are connected to an MLP inferring the labels. The SL is applied to the outputs of the two networks. Like DPL, the SL also exploits probabilistic-logic reasoning and knowledge compilation, but differs in that it cannot constrain predictions to satisfy the knowledge K at test time.

Logic Tensor Networks (LTN) [47, 48] softens the prior knowledge K using fuzzy logic to define a differentiable measure of label consistency and actively maximizes it during learning. At inference time [49], labels are predicted by first predicting the concepts by computing a MAP solution of $p_\theta(\mathbf{C} \mid \mathbf{x})$ and then combining it with the logic.

**Competitors.** We evaluate each architecture in isolation and in conjunction with bears and the following well-known calibration methods.

MC Dropout (MCDO) [50] consists in training a network with a dropout term and then averaging over an ensemble of concept extractors obtained by randomly deactivating neurons during inference;

The Laplace approximation (LA) [37] approximates the Bayesian posterior by placing a normal distribution around the trained concept extractor, applying a covariance proportional to the inverse of the Hessian matrix computed on the label loss;

Deep Ensembles (DE) [39], like bears, trains an ensemble of models (with different hyperparameters, like the random seed and learning rate, to ensure different optimization between models) under the same objective and a noise adver-

sarial term, but it does not contain any knowledge-unaware diversification penalty. After training, the mean concept extractor is given by the average of the ensemble members.

We also consider a Probabilistic Concept-bottleneck Models (PCBM) [51] backbone, an interpretable neural network architecture that outputs a normal distribution for each concept, implicitly improving uncertainty calibration. Concept probabilities are predicted by instantiating two sets of prototypical vectors, one for positive values of each concept $C_i$ and one for the negative values, similar to concept embedding models [46]. The sigmoid of the relative distance of the network embedding to each of these prototypes then gives the concept probability $p_\theta(\mathbf{C} \mid \mathbf{x})$. Hyperparameters and further implementation details are reported in Appendix A.

For both labels and concepts, we report – averaged over 5 seeds – both *prediction quality* and *calibration*, measured in terms of $F_1$ score (or accuracy) and Expected Calibration Error (ECE), respectively. A higher ECE (reported explicitly in Eq. (11)) indicates that a model is overconfident in giving wrong predictions, which we expect to be the case for methods not modeling uncertainty explicitly. The subscript $Y$ (resp. $C$) indicates we are measuring label (resp. concept) calibration error. See Appendix A.2 for definitions. We also report a runtime comparison in Appendix A.8.

**Data sets.** We consider two variants of MNIST addition [8], which requires predicting the sum of two MNIST [52] digits, except that only selected pairs of digits are observed during training. MNIST-Half includes only sums of digits 0 through 4 chosen so that only the semantics of the digit 0 can be unequivocally determined from data. The combinations include:

$$\begin{cases} \mathbf{0} + \mathbf{0} &= 0 \\ \mathbf{0} + \mathbf{1} &= 1 \end{cases} \quad \begin{cases} \mathbf{2} + \mathbf{3} &= 5 \\ \mathbf{2} + \mathbf{4} &= 6 \end{cases} \quad (9)$$

MNIST-Even-Odd is similar, except that it covers all digits; all in-distribution combinations are reported explicitly in Appendix A.4. It was introduced in [3] to evaluate the impact of RSs in DPL, SL, and LTN

Kandinsky is a variant of the Kandinsky Patterns task [53], where given three images containing three simple colored shapes each (*e.g.*, two red squares and a blue triangle) and a logical combination of rules like "the three objects have different shapes" or "they have the same color", the goal is to predict whether the third image satisfies the same rules as the first two. The example in Fig. 3 provides an idea of this task.

BDD-OIA [1, 2] is a real-world multi-label prediction task in which the goal is to predict what actions, out of {forward, stop, right, left}, are safe based on objects (like pedestrians and red lights) that are visible in a given dashcam image and prior knowledge akin to that in Example 1. The data set comprises 21 concepts, indicating the

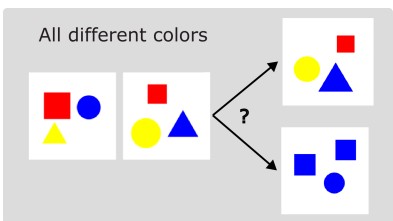

Figure 3: An example of a test sample for the Kandinsky task. At inference time, the NeSy model has to choose according to the previous two images the third that completes the *pattern*. The model computes a series of predicates for each image, like same_colors, same_shapes. In the running example, the first two images have different colors, so the model should pick the first option.

Table 1: **bears dramatically improves RS-awareness across the board**. All tested architectures achieve substantially better concept-level ECE and out-of-distribution label-level ECE, with comparable in-distribution label-level ECE. Results for MNIST-Half are shown. MNIST-Even-Odd shows a similar trend (see Table 11 in the Appendix).

| METHOD | $\text{ECE}_Y$ | $\text{ECE}_C$ | $\text{ECE}_{Y_{ood}}$ | $\text{ECE}_{C_{ood}}$ |
|---|---|---|---|---|
| DPL | $0.02 \pm 0.01$ | $0.69 \pm 0.01$ | $0.92 \pm 0.01$ | $0.87 \pm 0.01$ |
| + MCDO | $0.02 \pm 0.01$ | $0.69 \pm 0.01$ | $0.91 \pm 0.01$ | $0.86 \pm 0.01$ |
| + LA | $0.06 \pm 0.01$ | $0.65 \pm 0.01$ | $0.87 \pm 0.01$ | $0.82 \pm 0.01$ |
| + PCBM | $0.07 \pm 0.08$ | $0.64 \pm 0.08$ | $0.86 \pm 0.08$ | $0.80 \pm 0.08$ |
| + DE | $\mathbf{0.01 \pm 0.01}$ | $0.64 \pm 0.01$ | $0.83 \pm 0.13$ | $0.77 \pm 0.13$ |
| + bears | $0.09 \pm 0.02$ | $\mathbf{0.37 \pm 0.01}$ | $\mathbf{0.39 \pm 0.03}$ | $\mathbf{0.38 \pm 0.02}$ |
| SL | $\mathbf{0.01 \pm 0.01}$ | $0.71 \pm 0.01$ | $0.95 \pm 0.01$ | $0.88 \pm 0.01$ |
| + MCDO | $\mathbf{0.01 \pm 0.01}$ | $0.70 \pm 0.01$ | $0.92 \pm 0.01$ | $0.88 \pm 0.01$ |
| + LA | $0.06 \pm 0.01$ | $0.59 \pm 0.02$ | $\mathbf{0.75 \pm 0.01}$ | $0.75 \pm 0.02$ |
| + PCBM | $\mathbf{0.01 \pm 0.01}$ | $0.70 \pm 0.01$ | $0.91 \pm 0.01$ | $0.88 \pm 0.01$ |
| + DE | $\mathbf{0.01 \pm 0.01}$ | $0.64 \pm 0.08$ | $0.87 \pm 0.05$ | $0.78 \pm 0.13$ |
| + bears | $\mathbf{0.01 \pm 0.01}$ | $\mathbf{0.38 \pm 0.01}$ | $\mathbf{0.75 \pm 0.01}$ | $\mathbf{0.37 \pm 0.03}$ |
| LTN | $0.02 \pm 0.01$ | $0.70 \pm 0.01$ | $0.94 \pm 0.01$ | $0.87 \pm 0.01$ |
| + MCDO | $\mathbf{0.01 \pm 0.01}$ | $0.69 \pm 0.01$ | $0.93 \pm 0.01$ | $0.87 \pm 0.01$ |
| + LA | $0.14 \pm 0.02$ | $0.55 \pm 0.02$ | $0.79 \pm 0.02$ | $0.73 \pm 0.02$ |
| + PCBM | $\mathbf{0.01 \pm 0.01}$ | $0.69 \pm 0.01$ | $0.94 \pm 0.01$ | $0.86 \pm 0.01$ |
| + DE | $\mathbf{0.01 \pm 0.01}$ | $0.69 \pm 0.01$ | $0.94 \pm 0.01$ | $0.87 \pm 0.01$ |
| + bears | $0.06 \pm 0.01$ | $\mathbf{0.36 \pm 0.01}$ | $\mathbf{0.36 \pm 0.01}$ | $\mathbf{0.32 \pm 0.01}$ |

presence of pedestrians, red and green traffic lights, and other kinds of objects common in road traffic. The rules prevent the model from predicting actions whenever these are unsafe due to, *e.g.*, presence of obstacles in the corresponding direction. The forward and stop actions do share concepts, *e.g.*,

$$\begin{cases} \text{red\_light} \vee \text{stop\_sign} \vee \text{obstacle} \Rightarrow \text{stop} \\ \text{stop} \Rightarrow \neg \text{move\_forward} \end{cases}$$

See Appendix A.4 for a longer description of all data sets.

**Q1: RSs make NeSy predictors overconfident.** Table 1 lists the label and concept ECE of all competitors on MNIST-Half, measured both in-distribution (sums in the training set) and out-of-distribution (all other sums). The label and concept accuracy are reported in the appendix (Table 10) due to space constraints. Overall, all NeSy pre-

Table 2: **`bears` dramatically improves RS-awareness in the real-world.** Results on `BDD-OIA` with `DPL` show substantial ECE improvements both jointly ($\text{mECE}_C$) and for different classes of concepts (F=`forward`, S=`stop`, R=`turn right`, L=`turn left`).

| | $\text{mECE}_C$ | $\text{ECE}_C(F, S)$ | $\text{ECE}_C(R)$ | $\text{ECE}_C(L)$ |
|---|---|---|---|---|
| `DPL` | $0.84 \pm 0.01$ | $0.75 \pm 0.17$ | $0.79 \pm 0.05$ | $0.59 \pm 0.32$ |
| `+ MCDO` | $0.83 \pm 0.01$ | $0.72 \pm 0.19$ | $0.76 \pm 0.08$ | $0.55 \pm 0.33$ |
| `+ LA` | $0.85 \pm 0.01$ | $0.84 \pm 0.10$ | $0.87 \pm 0.04$ | $0.67 \pm 0.19$ |
| `+ PCBM` | $0.68 \pm 0.01$ | $0.26 \pm 0.01$ | $0.26 \pm 0.02$ | $0.11 \pm 0.02$ |
| `+ DE` | $0.79 \pm 0.01$ | $0.62 \pm 0.03$ | $0.71 \pm 0.10$ | $0.37 \pm 0.12$ |
| `+ bears` | $\mathbf{0.58 \pm 0.01}$ | $\mathbf{0.14 \pm 0.01}$ | $\mathbf{0.10 \pm 0.01}$ | $\mathbf{0.02 \pm 0.01}$ |

dictors achieve high label accuracy ($\geq 90\%$) but fare poorly in terms of concept accuracy (approx. $43\%$ for `DPL` and `SL`, and `LTN`), meaning they are affected by RSs, as expected. Our results also show that they are *not RS-aware*, as they are very confident about their concept predictions ($\text{ECE}_C$ of approx. $69\%$ for `DPL`, $71\%$ for `SL`, and $70\%$ fort `LTN`). Moreover, the label predictions are well calibrated ($\text{ECE}_Y$ is approx. $2\%$ for `DPL` and `LTN`, $1\%$ for `SL`), meaning that label uncertainty is not a useful indicator of RSs. In general, models performance worsens out-of-distribution in terms of label accuracy (barely above $0$ for all models) and label and concept calibration (ECE around $90\%$), despite concept accuracy remaining roughly stable (about $40\%$). The results for `MNIST-Even-Odd` follow a similar trend, cf. Table 11 in the appendix.

As for `BDD-OIA`, we only evaluate `DPL` as it is the only model that guarantees predictions comply with the safety constraints out of the ones we consider. The results in Table 2 and Table 12 show that `DPL` achieves good label accuracy ($72\%$ macro $F_1$) in this challenging task by leveraging poor concepts ($34\%$ macro $F_1$) with high confidence ($\text{mECE}_C \approx 84\%$). This supports our claim that NeSy architectures are not RS-aware.

**Q2: Combining NeSy predictors with `bears` dramatically improves RS-awareness in all data sets while retaining the same prediction accuracy.** For `MNIST-Half` (Table 1, Table 10), `bears` shrinks the concept ECE from $69\%$ to $37\%$ for `DPL`, from $71\%$ to $38\%$ for `SL`, and from $70\%$ to $36\%$ for `LTN` in-distribution. The out-of-distribution improvement even more substantial, as `bears` improves both concept calibration (`DPL`: $87\% \to 38\%$, `SL`: $88\% \to 37\%$, `LTN`: $87\% \to 32\%$) *and* label calibration (`DPL`: $92\% \to 39\%$, `SL`: $95\% \to 75\%$, `LTN`: $94\% \to 36\%$). No competitor comes close. The runner-up, `LA`, improves concept calibration on average by $10.5\%$ and at best by $15\%$ (for `LTN` in-distribution), while `bears` averages $42.3\%$ and up to $55\%$ (for `LTN` out-of-distribution). Fig. 4 shows that `bears` correctly assigns high uncertainty to all digits but the zero, which is the only one not affected by RSs, while all competitors are largely overconfident on these digits.

In `BDD-OIA` the trend is largely the same: `bears` im-

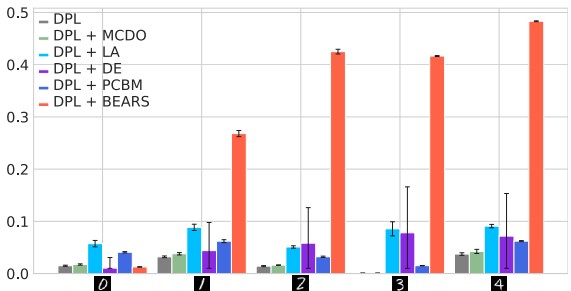

Figure 4: **Per-concept entropy shows `bears` is more uncertain about concepts affected by RS** on `MNIST-Half` compared to regular `DPL` and alternative uncertainty calibration methods. `SL` and `LTN` show similar trends, see Appendix C. Importantly, these improvements do not require concept annotations.

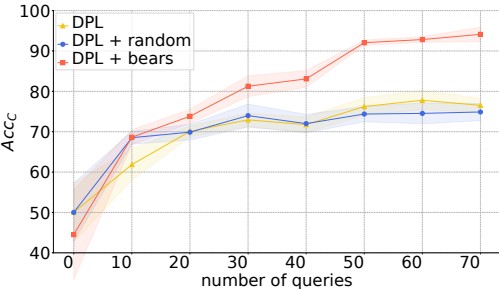

Figure 5: **`bears` allows selecting informative concept annotations faster.** A substantial improvement in concept accuracy is achieved by performing active learning guided by RS-aware concept uncertainty (`DPL +bears`) with respect to plain concept uncertainty (`DPL`) and random selection.

proves the test set concept ECE for all concepts jointly ($\text{mECE}_C$) from $84\%$ to $58\%$ ($-26\%$). The improvement becomes even clearer if we group the various concepts based on what actions they entail: concepts for `forward`/`stop` improve by $-61\%$, those for `right` by $-69\%$, and those for `left` by $-57\%$. Here, `LA` performs quite poorly (in fact, it yields *worsen* calibration), and the runner-up `PCBM`, which fares well ($-16\% \text{ mECE}_C$), is also substantially worse than `bears`. Finally, we note that, despite their similarities, `DE` underperforms overall, showcasing the importance of our knowledge-aware ensemble diversification strategy.

**Q3: `bears` allows for selecting better concept annotations.** Fig. 5 reports the results in terms of concept accuracy on the `Kandinsky` dataset when using an active learning strategy to acquire concept supervision. Results are obtained by pre-training `DPL` with 10 examples of red squares, and selecting additional objects for supervision based on their concept uncertainty. Results show that standard `DPL` has the same behaviour as a random sampling strategy, likely because of its poor estimation of concept uncertainty. On the other hand, `DPL` with `bears` manages to substantially outperform both alternatives in improving concept accu-

racy, achieving an accuracy of more than 90% with just 50 queries, while the other strategies level off at around 75% accuracy. Note that because of the presence of reasoning shortcuts, all models achieve high label-level accuracy regardless of their concept-level accuracy. See Appendix A.7 for the details.

# 5 RELATED WORK

**Neuro-Symbolic Integration.** NeSy AI [20, 21] spans a broad family of models and tasks – both discriminative and generative – involving perception and reasoning [5, 6, 12, 54, 55]. Given discrete reasoning is not differentiable, NeSy architectures support end-to-end training either by imbuing the prior knowledge with probabilistic [56, 8, 57, 58, 59, 10, 23, 60] or fuzzy [61, 62, 47] semantics, by implementing reasoning in embedding space [63], or through a combination thereof [64]. Another difference is whether they encourage [7, 65, 64] vs. guarantee [8, 9, 10, 11, 66] predictions to be consistent with the knowledge. Despite their differences, all NeSy approaches can be prone to RSs, which occur whenever prior knowledge – including label supervision – is insufficient to pin down the correct concept semantics.

**Dealing with Reasoning Shortcuts.** Existing works on RSs focus on unsupervised mitigation, often by discouraging learned concepts from collapsing onto each other. Examples include using a batch-wise entropy loss [67], a reconstruction loss [3], a bottleneck maximization approach [68], and encouraging constraint satisfaction via non-trivial assignments [13]. Our work builds on the insights from Marconato *et al.* [3], who recently showed that unsupervised mitigation only works in specific cases, and that only expensive strategies – like multi-task learning and dense annotations [15] – can provably avoid RSs in all cases. Other works based on *abductive learning* [69, 70] constitute promising avenues for lessening the impact of RSs.

Our key contribution is that of switching focus from mitigation to awareness, which – as we show – *can* be achieved in an unsupervised manner. In this sense, bears is closely related to unsupervised mitigation heuristics [67, 68, 13], but differs in the goal (awareness vs. mitigation). bears specifically averages over neural networks that capture conflicting RSs to achieve knowledge-dependent uncertainty calibration. It is also related to the neuro-symbolic entropy of [22], which, however, *minimizes* instead of maximizing the entropy of the NeSy predictor, and as such it can exacerbate the negative effects of RSs. In our analysis, we characterize awareness of RSs in the limit case of infinite data, and future work can provide statistical guarantees about the uncertainty of concepts and final task performance, for example adapting results from [70, 71].

**Uncertainty calibration in deep learning.** Overconfidence of deep learning models is a well-known issue [72]. Many strategies for reducing overconfidence of label predictions exist [73, 74, 75, 76, 77], many of which based on Bayesian techniques [50, 37, 39]. Our experiments show that applying them to NeSy predictors fails to produce RS-aware models, whereas bears succeeds. Techniques from concept-based models for imbuing concepts with probabilistic semantics [51, 78] also can improve calibration, but underperform in our experiments compared to bears.

# 6 CONCLUSION

NeSy models tend to be unaware of RSs affecting them, hindering reliability. We address this by introducing bears, which encourages NeSy models to be more uncertain about concepts affected by RSs, enabling users to identify and distrust bad concepts. bears vastly improves RS-awareness compared to NeSy baselines and state-of-the-art calibration methods while retaining high prediction accuracy, and lowers the cost of supervised mitigation via uncertainty-based active learning of dense annotations. In future work, we will explore richer knowledge acquisition strategies to encourage RS-awareness and reduce their impact, and look into leveraging causal representation learning [29, 32, 79] to define provably effective mitigation strategies. Furthermore, concurrent work proves that the NeSy predictors studied in this paper are fundamentally limited in expressing uncertainty, and that this can be overcome by increasing model expressivity using ensembling [80]. We will explore this relation with bears.

## Author Contributions

E.v.K. conceived the idea of tackling RSs using uncertainty. All authors contributed to the conceptualization and the writing. E.M. and S.B. implemented the code and carried out the empirical evaluation. E.M. and E.v.K. analyzed the theoretical backing of our approach. S.T., A.V., and A.P. supervised the work. A.V. and A.P. managed fund acquisition.

## Acknowledgements

The authors are grateful to Zhe Zeng for useful discussion. Funded by the European Union. The views and opinions expressed are however those of the author(s) only and do not necessarily reflect those of the European Union or the European Health and Digital Executive Agency (HaDEA). Neither the European Union nor the granting authority can be held responsible for them. Grant Agreement no. 101120763 - TANGO. AV is supported by the "UNREAL: Unified Reasoning Layer for Trustworthy ML" project (EP/Y023838/1) selected by the ERC and funded by UKRI EPSRC. Emile van Krieken was funded by ELIAI (The Edinburgh Laboratory for Integrated Artificial Intelligence), EPSRC (grant no.

EP/W002876/1).

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

# A IMPLEMENTATION DETAILS

In this Section, we provide additional details about all metrics, datasets and models useful for reproducibility.

## A.1 IMPLEMENTATION

All the experiments are implemented using Python 3.8 and Pytorch 1.13 and run on one A100 GPU. The implementations of `DPL`, `SL`, and `LTN` were taken verbatim from [3]. We implemented `MCDO` and `DE` by adapting the code to capture the original algorithms [50, 39]. For `PCBM`, we followed the original paper [51]. For `LA`, we adapted the original library from `laplace-torch` [37]. In our experiments, we computed the Laplace approximation on the second last layer, mapping the embeddings to the concept layer. The images for `Kandinsky` patterns were synthetically originated from the resource provided in [53].

## A.2 METRICS

For all datasets, we evaluate the predictions on the labels by measuring the accuracy and the $F_1$-score with macro average. We assess calibration using the Expected Calibration Error (ECE), which measures how accurately the model-predicted probabilities align with actual data likelihood. Specifically, for a given label $y_i \in \mathbb{N}$, the $\text{ECE}_Y(i)$ for each label error is evaluated as:

$$\text{ECE}_Y(i) = \sum_{\ell=1}^{M} \frac{|B_\ell|}{n} |\text{Acc}_Y(B_\ell) - \text{Conf}_Y(B_\ell)|, \quad \forall i \in [m] \tag{10}$$

where $M$ is the number of bins, $B_m$ represent the $m$-th bin and $\text{Conf}_Y$ denotes the predicted probability. Essentially, the predicted probabilities are categorized into intervals, denoted as bins. Each data point is assigned to a bin based on its predicted probability. Within each bin, the average predicted probability and accuracy are computed. Ultimately, the ECE value is obtained by summing the averages of absolute differences between predicted probabilities and accuracies. Similarly, we evaluate $\text{ECE}_C(j)$ as:

$$\text{ECE}_C(j) = \sum_{\ell=1}^{M} \frac{|B_\ell|}{n} |\text{Acc}_C(B_\ell) - \text{Conf}_C(B_\ell)|, \quad \forall j \in [k] \tag{11}$$

In `MNIST-Half` and `MNIST-Even-Odd` we use the very same network to extract the first and second digits, and similarly in `Kandinsky` for extracting the color and shape of each object. For this reason, $\text{ECE}_C$ was evaluated by stacking the concepts predicted by the architecture for each object. $\text{ECE}_Y$ was evaluated on the final predictions.

In contrast, `BDD-OIA` images involve multiple concepts and multiple labels. In this case, we adopted a softer approach, specifically we averaged over the performances on each separate component:

$$\text{mECE}_Y = \frac{1}{m} \sum_{i=1}^{m} \text{ECE}_Y(i) \tag{12}$$

$$\text{mECE}_C = \frac{1}{k} \sum_{i=1}^{k} \text{ECE}_C(i) \tag{13}$$

where $l$ and $k$ are the numbers of labels and concepts, respectively.

In `MNIST-Addition` and its variations, we evaluate all metrics both in-distribution and out-of-distribution. In `Kandinsky`, labels and concepts are both balanced, so we report accuracy for both. In `BDD-OIA` the data is not as balanced, so we report the mean-$F_1$, score as in [2, 3], that is, we first compute the $F_1$-score for each action and then average them:

$$\text{mF}_1(Y) = \frac{F_1(\text{forward}) + F_1(\text{stop}) + F_1(\text{left}) + F_1(\text{right})}{4} \tag{14}$$

For all datasets, to measure uncertainty concept-wise for a specific model $p_\theta$, we rely on the one-vs-all entropy. We evaluate the average entropy of $p_\theta(C = c \mid \mathbf{x})$ as:

$$H_{OVA}(p_\theta(C = c|X)) = -\frac{1}{|\mathcal{D}|} \sum_{\mathbf{x}\in\mathcal{D}} \big[ p_\theta(C = c|\mathbf{x}) \log(p_\theta(C = c|\mathbf{x})) + (1 - p_\theta(C = c|\mathbf{x})) \log(1 - p_\theta(C = c|\mathbf{x})) \big] \quad (15)$$

## A.3 `BEARS` IMPLEMENTATION

Implementation-wise, `bears` is an extension of `DE` with a new concept-level repulsive term. In short, `bears` works as follows. For each new model $\theta_t$ to be added to the ensemble, we compute the following loss by considering all other members in $\boldsymbol{\theta} = \{\theta_j\}_{j=1}^{t-1}$:

$$\max_{\theta_t} \frac{1}{|\mathcal{D}|} \sum_{(\mathbf{x},\mathbf{y})\in\mathcal{D}} \Big[ \log p_{\theta_t}(\mathbf{y} \mid \mathbf{x}; \mathsf{K}) + \gamma_1 \mathsf{KL}\big(p_{\theta_t}(\mathbf{C} \mid \mathbf{x}) \,\|\, \frac{1}{t}\sum_{j=1}^{t} p_{\theta_j}(\mathbf{C} \mid \mathbf{x})\big) + \gamma_2 H\big(p_{\theta_t}(\mathbf{C} \mid \mathbf{x})\big) \Big] \quad (16)$$

We can analyze further the expression of the KL divergence to express it differently:

$$\mathsf{KL}\big(p_{\theta_t}(\mathbf{C} \mid \mathbf{x}) \,\|\, \frac{1}{t}\sum_{j=1}^{t} p_{\theta_j}(\mathbf{C} \mid \mathbf{x})\big) = \sum_{\mathbf{c}\in\{0,1\}^k} p_{\theta_t}(\mathbf{c} \mid \mathbf{x}) \log \frac{p_{\theta_t}(\mathbf{c} \mid \mathbf{x})}{\frac{1}{t}\sum_{j=1}^{t-1} p_{\theta_j}(\mathbf{c} \mid \mathbf{x}) + \frac{1}{t} p_{\theta_t}(\mathbf{c} \mid \mathbf{x})} \quad (17)$$

$$= -\sum_{\mathbf{c}\in\{0,1\}^k} p_{\theta_t}(\mathbf{c} \mid \mathbf{x}) \log \frac{1}{t} \cdot \frac{\sum_{j=1}^{t-1} p_{\theta_j}(\mathbf{c} \mid \mathbf{x}) + p_{\theta_t}(\mathbf{c} \mid \mathbf{x})}{p_{\theta_t}(\mathbf{c} \mid \mathbf{x})} \quad (18)$$

$$= \sum_{\mathbf{c}\in\{0,1\}^k} p_{\theta_t}(\mathbf{c} \mid \mathbf{x}) \log t - \log\left[ 1 + (t-1) \cdot \sum_{j=1}^{t-1} \frac{1}{t-1} \frac{p_{\theta_j}(\mathbf{c} \mid \mathbf{x})}{p_{\theta_t}(\mathbf{c} \mid \mathbf{x})} \right] \quad (19)$$

$$= \log t - \sum_{\mathbf{c}\in\{0,1\}^k} p_{\theta_t}(\mathbf{c} \mid \mathbf{x}) \log\left[ 1 + (t-1) \cdot \frac{p_{rest}(\mathbf{c} \mid \mathbf{x})}{p_{\theta_t}(\mathbf{c} \mid \mathbf{x})} \right] \quad (20)$$

where in the second line we introduced a minus sign to flip the term in the logarithm, in the third line we have taken out $p_{\theta_t}(\mathbf{c} \mid \mathbf{x})$ from the numerator and multiplied and divided the remaining terms for $(t-1)$, and in the last line we denoted with $p_{rest}(\mathbf{c} \mid \mathbf{x})$ the average on the members of the ensemble up to $t-1$.

In general, the KL divergence is unbounded from above but since the same distribution $p_{\theta_t}(\mathbf{C} \mid \mathbf{x})$ appears from both sides this gives an upper-bound. Notice that, since the KL is always greater or equal than zero we have that:

$$0 \le \mathsf{KL}\big(p_{\theta_t}(\mathbf{C} \mid \mathbf{x}) \,\|\, \frac{1}{t}\sum_{j=1}^{t} p_{\theta_j}(\mathbf{C} \mid \mathbf{x})\big) \le \log t \quad (21)$$

Following, we consider the composite expression with the term proportional to the entropy on the concepts $p_\theta(\mathbf{C} \mid \mathbf{x})$, without accounting for the $\log t$ term. In our implementation, we minimize the term:

$$\min_\theta \frac{1}{|\mathcal{D}|} \sum_{(\mathbf{x},\mathbf{y})\in\mathcal{D}} p_\theta(\mathbf{y} \mid \mathbf{x}; \mathsf{K}) + \frac{\gamma_1}{\log t} \sum_{\mathbf{c}\in\{0,1\}^k} p_{\theta_t}(\mathbf{c} \mid \mathbf{x}) \log\left[ 1 + (t-1) \cdot \frac{p_{rest}(\mathbf{c} \mid \mathbf{x})}{p_\theta(\mathbf{c} \mid \mathbf{x})} \right]$$
$$+ \gamma_2\left( 1 - \frac{H(p_{\theta_t}(\mathbf{C} \mid \mathbf{x}))}{k \log 2} \right) \quad (22)$$

for each new member of the ensemble, where we divided the KL term by $\log t$ to ensure its normalization, and we normalized the entropy for the maximal value $k \log 2$. The pseudo-code of `bears` is shown in Algorithm 1.

## A.4 DATASETS DETAILS

In our experiments, when possible, we processed different digits and objects with the same neural network. This happens in both `MNIST-Addition` tasks and `Kandinsky`, whereas for `BDD-OIA` this choice is not available.

**Algorithm 1** `bears`

---

1: **procedure** BEARS($n, seeds, \gamma_1, \gamma_2, epochs, train\_loader$)
2:     Initialize empty `ensemble`
3:     **for** $i = 1 \ldots n$ **do**
4:         $seed \leftarrow seeds[i]$ # Set seed using `seeds[i]`
5:         $model = $ `get_neq_model`($seed$) # Create a new ANN model from the seed
6:         **for** $e = 1, \ldots,$ `epochs` **do**
7:             **for** data $(x, y)$ **in** `train_loader` **do**
8:                 $\hat{y}, pcx = model(x)$ # Compute $\hat{y}$ and $p(c \mid x)$
9:                 $loss = \mathrm{C}(y, \hat{y})$ # Calculate the loss in classification for the NeSy model
10:                 **if** $i > 0$ **then**
11:                     $\overline{pcx} = \mathrm{mean}(pcx)$ # Compute the ensemble average $\overline{p(c|x)}$
12:                     $loss = loss + \gamma_1 \ \mathrm{KL}(pcx \ || \ \overline{pcx}) + \gamma_2 \ H(pcx)$ # Update loss with the KL term and entropy penalty
13:                 $loss.\mathrm{backprop}()$ # Backpropagate the loss and update model parameters
14:         $ensemble[i] \leftarrow model$ # Add `model` to `ensemble`
15:     **return** `ensemble`

---

### A.4.1 `MNIST-Even-Odd`

As done in [3], we considered the `MNIST-Even-Odd` dataset, initially introduced in [15]. This variant of `MNIST-Addition` has only a few specific combinations of digits, containing either only even or only odd digits:

$$
\begin{cases}
\boxed{0} + \boxed{6} & = 6 \\
\boxed{2} + \boxed{8} & = 10 \\
\boxed{4} + \boxed{6} & = 10 \\
\boxed{4} + \boxed{8} & = 12
\end{cases}
\quad \wedge \quad
\begin{cases}
\boxed{1} + \boxed{5} & = 6 \\
\boxed{3} + \boxed{7} & = 10 \\
\boxed{1} + \boxed{9} & = 10 \\
\boxed{3} + \boxed{9} & = 12
\end{cases}
\tag{23}
$$

`MNIST-Even-Odd` consists of a total of 6720 fully annotated samples in the training set, 1920 samples in the validation set, and 960 samples in the in-distribution test set. Additionally, there are 5040 samples in the out-of-distribution test dataset comprising all other sums that are not observed during training.

**Reasoning Shortcuts:** As described in [3], the number of deterministic RSs can be calculated by finding the integer values for the digits $\boxed{0}, \ldots, \boxed{9}$ that solve the above linear system. In total, it was shown that the number of deterministic RSs amounts to 49.

### A.4.2 `MNIST-Half`

This dataset constitutes a biased version of `MNIST-Addition`, including only half of the digits, specifically, those ranging from 0 to 4. Moreover, we selected the following combinations of digits:

$$
\begin{cases}
\boxed{0} + \boxed{0} & = 0 \\
\boxed{0} + \boxed{1} & = 1 \\
\boxed{2} + \boxed{3} & = 5 \\
\boxed{2} + \boxed{4} & = 6
\end{cases}
\tag{24}
$$

This allows introducing several RSs for the system. Unlike `MNIST-Even-Odd`, two digits are not affected by reasoning shortcuts: namely 0 and 1. The remaining, 2, 3, and 4 can be predicted differently, as shown below.

In total, `MNIST-Half` comprises 2940 fully annotated samples in the training set, 840 samples in the validation set, 420 samples in the test set, and an additional 1080 samples in the out-of-distribution test dataset. These only comprise the remaining sums with these digits, like $\boxed{1} + \boxed{3} = 4$.

**Reasoning shortcuts:** We identify all the possible RSs empirically, since the system of observed sums can be written as a linear system from Eq. (24). There are in total three possible optimal solutions, of which two are reasoning shortcuts. Explicitly:

$$\boxed{0} \mapsto 0, \boxed{1} \mapsto 1, \boxed{2} \mapsto 2, \boxed{3} \mapsto 3, \boxed{4} \mapsto 4$$
$$\vee$$
$$\boxed{0} \mapsto 0, \boxed{1} \mapsto 1, \boxed{2} \mapsto 3, \boxed{3} \mapsto 2, \boxed{4} \mapsto 3, \tag{25}$$
$$\vee$$
$$\boxed{0} \mapsto 0, \boxed{1} \mapsto 1, \boxed{2} \mapsto 4, \boxed{3} \mapsto 1, \boxed{4} \mapsto 2,$$

### A.4.3 `Kandinsky`

This dataset, introduced in [53], consists of visual patterns inspired by the artistic works of Wassily Kandinsky. These patterns are made of geometric figures, with several features. In our experiment, we propose a variant of `Kandinsky` where each image has a fixed number of figures, and the associated concepts are shape and color. In total, each object can take one among three possible colors (`red`, `blue`, `yellow`) and one among three possible shapes (`square`, `circle`, `triangle`).

We propose our `Kandinsky` variant for an active learning setup resembling an IQ test for machines. The task is to predict the pattern of a third image given two images sharing a common pattern.

Formally, let $x$ be an object in the figure, $S(x)$ the shape of $x$, and $C(x)$ its color. Let the image be denoted as $Figure$. In total, each figure contains three objects with possibly different colors and shapes. To enhance the clarity and conciseness of our logical expressions, we introduce the following shorthand predicates:

$$
\begin{cases}
\texttt{diff\_s}(Figure) \equiv \forall x, y \in Figure : (x \neq y \rightarrow \neg(S(x) = S(y))) \\
\texttt{diff\_c}(Figure) \equiv \forall x, y \in Figure : (x \neq y \rightarrow \neg(C(x) = C(y))) \\
\texttt{same\_s}(Figure) \equiv \forall x, y \in Figure : (S(x) = S(y)) \\
\texttt{same\_c}(Figure) \equiv \forall x, y \in Figure : (C(x) = C(y)) \\
\texttt{pair\_c}(Figure) \equiv \neg \texttt{same\_c}(Figure) \wedge \neg \texttt{diff\_c}(Figure) \\
\texttt{pair\_s}(Figure) \equiv \neg \texttt{same\_s}(Figure) \wedge \neg \texttt{diff\_s}(Figure)
\end{cases}
\tag{26}
$$

Let $Sample$ represent a training sample consisting of two figures for the sake of simplicity; the extension to more figures is trivial. The final logic statement, that determines the model output, is:

$$
\begin{aligned}
&\texttt{shared\_pattern} \Rightarrow \forall f_1, f_2 \in Sample : \\
&(\texttt{same\_c}(f_1) \wedge \texttt{same\_c}(f_2)) \vee (\texttt{pair\_c}(f_1) \wedge \texttt{pair\_c}(f_2)) \vee (\texttt{diff\_c}(f_1) \wedge \texttt{diff\_c}(f_2)) \\
&\qquad\qquad\qquad\qquad \vee \\
&(\texttt{same\_s}(f_1) \wedge \texttt{same\_s}(f_2)) \vee (\texttt{pair\_s}(f_1) \wedge \texttt{pair\_s}(f_2)) \vee (\texttt{diff\_s}(f_1) \wedge \texttt{diff\_s}(f_2))
\end{aligned}
\tag{27}
$$

Our `Kandinsky` dataset version comprises 4k examples in training, 1k in validation, and 1k in test.

We create our dataset to include a balanced number of positive and negative examples. Positive examples consist of three images sharing the same pattern, while in negative examples the third image does not match the pattern which is shared by the first two images. The order of examples does not introduce bias into the neural network learning procedure, as the network treats each figure independently.

**Preprocessing:** When processing an entire figure at once, we empirically observed that the model faces challenges in achieving satisfactory accuracy. Consequently, we opted to process one object at a time. Therefore, we employed a simplified version of the dataset that comprises of rescaled objects manually extracted via bounding boxes. Thus, each example of the dataset consists of 9 objects, namely 3 objects for each figure, ordered based on their distance from the origin of the figure.

**Reasoning shortcuts:** The knowledge we build for `Kandinsky` admits many RSs. As there are no constraints on specific colors or shapes, in principle, each permutation of colors and shapes can achieve perfect accuracy. Furthermore, the logic is symmetrical; hence, the concepts of colors and shapes could be swapped. Working on this dataset, we have observed various RSs. An example is illustrated below:

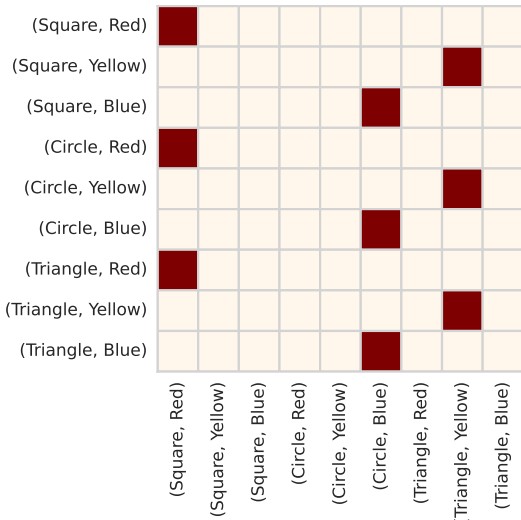

Figure 6: This plot shows an example of a RS in the `Kandinsky` task. The model achieves perfect accuracy by predicting shapes based on their colors. In this scenario, all red objects are correctly identified as squares, blue ones as circles, and yellow ones as triangles.

#### A.4.4 `BDD-OIA`

This dataset is made of frames retrieved from driving scene videos for autonomous driving predictions [1]. Each frame is labeled with four binary actions (`move_forward`, `stop`, `turn_left`, `turn_right`). Scenes are annotated with 21 binary concepts, providing explanations for the chosen actions. The training set includes 16k fully labeled frames, while the validation and test sets have 2k and 4.5k annotated data, respectively.

The prior knowledge employed is the same as in [3]. We report it here for the sake of completeness. For the `move_forward` and `stop` move, the rules are:

$$
\begin{cases}
\texttt{red\_light} \Rightarrow \neg\texttt{green\_light} \\
\texttt{obstacle} = \texttt{car} \vee \texttt{person} \vee \texttt{rider} \vee \texttt{other\_obstacle} \\
\texttt{road\_clear} \iff \neg\texttt{obstacle} \\
\texttt{green\_light} \vee \texttt{follow} \vee \texttt{clear} \Rightarrow \texttt{move\_forward} \\
\texttt{red\_light} \vee \texttt{stop\_sign} \vee \texttt{obstacle} \Rightarrow \texttt{stop} \\
\texttt{stop} \Rightarrow \neg\texttt{move\_forward}
\end{cases}
\tag{28}
$$

While for the `turn_left` and the `turn_right` action, the rules are:

$$
\begin{cases}
\texttt{can\_turn} = \texttt{left\_lane} \vee \texttt{left\_green\_lane} \vee \texttt{left\_follow} \\
\texttt{cannot\_turn} = \texttt{no\_left\_lane} \vee \texttt{left\_obstacle} \vee \texttt{left\_solid\_line} \\
\texttt{can\_turn} \wedge \neg\texttt{cannot\_turn} \Rightarrow \texttt{turn\_left}
\end{cases}
\tag{29}
$$

Moreover, for convenience in metric computations, we decided to group the actions into three classes of concepts. Specifically, we define $F - S$, which groups concepts concerning the `move_forward` and `stop` actions, $L$, which groups concepts concerning the `turn_left` action, and the $R$ group, which denotes the actions concerning the `turn_right` action. The classes are shown in Table 3.

### A.5 HYPERPARAMETERS AND MODEL SELECTION

In our work, we opted for the widely used Adam optimizer [81]. For `MNIST-Half` and `MNIST-Even-Odd`, the learning rate follows an exponential decay with $\gamma = 0.95$. Regarding `BDD-OIA`, the weight decay is $\omega = 1 \cdot 10^{-3}$ for all `DPL`

| Concept Class | Concepts |
|---|---|
| $F - S$ | green_light |
| | follow |
| | road_clear |
| | red_light |
| | traffic_sign |
| | car |
| | person |
| | rider |
| | other_obstacle |
| $L$ | left_lane |
| | left_green_light |
| | left_follow |
| | no_left_lane |
| | left_obstacle |
| | letf_solid_line |
| $R$ | right_lane |
| | right_green_light |
| | right_follow |
| | no_right_lane |
| | right_obstacle |
| | right_solid_line |

Table 3: Concept classes in BDD-OIA

variants, except for PCBM where we set it to 0.01. For the learning rate $\gamma$, it is set to 0.2 for DPL and its variants. However, we observed that a $\gamma = 1$ works best for PCBM since this model does not converge very early. In the active learning experiment on Kandinsky, we applied exponential decay with $\gamma = 0.9$.

To choose the hyperparameters, we conducted a grid search over a predefined set of values, and selected the best values based on both qualitative and quantitative results from a validation set. The learning rate for all experiments was fine-tuned within the range of $10^{-4}$ to $10^{-2}$. Specifically, for MNIST-Half, we set the learning rate to $5 \cdot 10^{-4}$ for DPL, and $1 \cdot 10^{-3}$ for SL, LTN, and PCBM. For Kandinsky, the learning rate was set to $1 \cdot 10^{-3}$. In the case of BDD-OIA , we explored a learning rate range between $10^{-4}$ and $10^{-2}$ and selected $10^{-3}$ for all the models.

Regarding batch sizes, we observed that 64 worked well for MNIST-Even-Odd and MNIST-Half, and 512 for BDD-OIA . For Kandinsky, a smaller batch size of 16 was chosen, as more frequent updates helped with model convergence.

Empirically, for bears, we discovered that optimizing $\gamma_1$ and $\gamma_2$ significantly influenced ensemble diversity, leading to different outcomes. Specifically, when these hyperparameters are much lower compared to the classification loss, the ensemble models tend to converge toward a single reasoning shortcut, reducing the impact of bears. Conversely, if these hyperparameters are bigger, the ensemble may consist of entirely different solutions, but potentially sub-optimal ones. These hyperparameters should be carefully tuned to strike a balance. We performed a grid search for both parameters over $\eta = \{0.1, 0.8, 0.5, 1, 2, 5, 10\}$ and selected the best values based on minimizing the classification objective and maximizing ensemble diversity.

For MNIST-Half and MNIST-Even-Odd, we observed that the impact of entropy is negligible. Consequently, we set $\gamma_2 = 0$ for all experiments. In contrast, relying solely on the KL term in BDD-OIA and Kandinsky does not effectively explore a consistent space of reasoning shortcuts. Thus, for bears, we set $\gamma_1 = 0.1$ for LTN and $\gamma_1 = 5$ for SL. For DPL, we set $\gamma_1 = 0.8$ for MNIST-Half and MNIST-Even-Odd, $\gamma_1 = 0.1$ and $\gamma_2 = 1$ for BDD-OIA , and $\gamma_1 = 0.01$ for Kandinsky.

Concerning the number of ensembles, a shared hyperparameter for DE and bears, we chose 5 for MNIST-Half, MNIST-Even-Odd, and Kandinsky, and 20 for BDD-OIA , considering the larger number of reasoning shortcuts in the latter. In BDD-OIA there $7^{21} \cdot 57^{114} \cdot 280^{280}$, compared to the 49 present in MNIST-Even-Odd [3].

Additionally, we observed that LTN behavior is quite unstable, resulting in sub-optimal models regardless of hyperparameter

Table 4: Encoder architecture for `MNIST-Half, MNIST-Even-Odd`.

| Input | Layer Type | Parameter | Activation |
|---|---|---|---|
| $(28, 56, 1)$ | Convolution | depth=32, kernel=4, stride=2, padding=1 | ReLU |
| $(32, 14, 28)$ | Dropout | $p = 0.5$ | |
| $(32, 14, 28)$ | Convolution | depth=64, kernel=4, stride=2, padding=1 | ReLU |
| $(64, 7, 14)$ | Dropout | $p = 0.5$ | |
| $(64, 7, 14)$ | Convolution | depth=128, kernel=4, stride=2, padding=1 | ReLU |
| $(128, 3, 7)$ | Flatten | | |
| $(2688)$ | Linear | dim=20, bias = True | |

Table 5: Encoder architecture for `Kandinsky`

| Input | Layer Type | Parameter | Activation |
|---|---|---|---|
| $(28, 28, 3)$ | Flatten | | |
| $(2352)$ | Linear | dim=256, bias=True | ReLU |
| $(256)$ | Dropout | $p = 0.5$ | |
| $(256)$ | Linear | dim=128, bias=True | ReLU |
| $(128)$ | Dropout | $p = 0.5$ | |
| $(128)$ | Linear | dim=8, bias = True | |

choices. To address this, we introduced an entropy penalization of $0.3$ to aid model convergence. The same approach was applied to `DPL` on `Kandinsky`, where this value was set to $0.2$.

Regarding the number of `LA` sampling and `MCDO`, we observed no big difference, thus we selected $30$ for our experiments.

Specifically for `LA`, we applied the Laplace approximation to the concept layer of the pre-trained frequentist model. For `MNIST-Half` and `MNIST-Even-Odd`, we used the Kronecker approximation of the Hessian matrix, but we could not use it for `BDD-OIA` due to excessive time and memory requirements. For `BDD-OIA`, we switched to the diagonal approximation.

For `PCBM`, the optimization involves the sum of two losses: a cross-entropy loss and a concept loss. The concept loss, denoted as $\mathcal{L}_{\text{concept}}$, is defined as $\mathcal{L}_{\text{concept}} = \mathcal{L}_{BCE} + \lambda_{\text{KL}}\mathcal{L}_{KL}$ [51]. Here, $\mathcal{L}_{BCE}$ represents the standard binary cross-entropy, and $\mathcal{L}_{KL}$ serves as a regularization term for the Gaussian distribution, defined as $\mathcal{L}_{KL} = \mathsf{KL}(N(\mu_c, \text{diag}(\sigma_c))||N(0, I))$. Since we lack concept supervision during training, the weight associated with the binary cross-entropy is set to $0$. The regularization term $\lambda_{kl}$ is maintained at $0.001$ for both examples, as setting it too high led to sub-optimal models in our specific context.

Finally, concerning the active learning example on `Kandinsky`, we found that to achieve optimal convergence while still learning concepts, effective parameters for the concept supervision loss are $25$ for `DPL` and $10$ for `DPL + bears`.

## A.6   ARCHITECTURES AND MODEL DETAILS

`MNIST-Addition`:   The architectures employed for `MNIST-Even-Odd` and `MNIST-Half` are essentially the one implemented in [3], outlined in Table 4. The only difference among the two datasets is the size of the bottleneck, which depends on the number of concepts. For `MNIST-Half`, the last layer dimension is 10, while for `MNIST-Even-Odd` is 20. For `PCBM`, the architecture is shown in Table 6. Additionally, for `SL` only, we introduced an MLP with a hidden size of 50 neurons. This MLP takes the logits of both concepts as input and processes them to produce the final label.

`BDD-OIA`:   Likewise for [3], `BDD-OIA` images have been preprocessed, as detailed in [2], employing a Faster-RCNN [82] pre-trained on MS-COCO and fine-tuned on BDD-100k, for initial preprocessing. Subsequently, we employ a pre-trained convolutional layer from [2] to extract linear features with a dimensionality of 2048. These linear features serve as inputs for the NeSy model, implemented with a fully-connected classifier network as outlined in Table 7 for `DPL`, and in Table 8 for `PCBM`.

Table 6: Encoder architecture for PCBM

| INPUT | LAYER TYPE | PARAMETER | ACTIVATION | NOTE |
|---|---|---|---|---|
| $(28, 56, 1)$ | Convolution | depth=32, kernel=4, stride=2, padding=1 | ReLU | |
| $(32, 14, 28)$ | Dropout | $p = 0.5$ | | |
| $(32, 14, 28)$ | Convolution | depth=64, kernel=4, stride=2, padding=1 | ReLU | |
| $(64, 7, 14)$ | Dropout | $p = 0.5$ | | |
| $(64, 7, 14)$ | Convolution | depth=128, kernel=4, stride=2, padding=1 | ReLU | |
| $(128, 3, 7)$ | Flatten | | | |
| $(2688)$ | Linear | dim=160, bias = True | | Head for $\mu$ |
| $(2688)$ | Linear | dim=160, bias = True | | Head for $\sigma$ |

Table 7: DPL architecture for BDD-OIA

| INPUT | LAYER TYPE | PARAMETER | ACTIVATION | NOTE |
|---|---|---|---|---|
| $(2048, 1)$ | Linear | dim=512, bias=True | ReLU | |
| $(512)$ | Dropout | | | |
| $(512)$ | Linear | dim=21, bias=True | | Head for move_forward action |
| $(512)$ | Linear | dim=12, bias=True | | Head for stop action |
| $(512)$ | Linear | dim=12, bias=True | | Head for turn_left action |
| $(512)$ | Linear | dim=12, bias=True | | Head for turn_right action |

Table 8: PCBM architecture for BDD-OIA

| INPUT | LAYER TYPE | PARAMETER | ACTIVATION | NOTE |
|---|---|---|---|---|
| $(2048, 1)$ | Linear | dim=336, bias=True | | Head for $\mu$ |
| $(2048, 1)$ | Linear | dim=336, bias=True | | Head for $\sigma$ |

Kandinsky: For Kandinsky, we chose to use an MLP-based encoder, as depicted in Table 5.

## A.7 ACTIVE LEARNING SETUP

The active learning setup proposed is based on Kandinsky. The examples consist of three figures, each composed of three objects. Each object is characterized by shape and color properties. In this setup, the model processes each object independently, producing a 6-dimensional vector that includes the one-hot encoding of shapes and colors, with each dimension representing one of the three shapes or colors. Overall, for each figure the model produces an 18-dimensional vector. The supervision is provided for a single object and consists of its shape and color.

To configure the experiment, we masked all the concepts in the training set, revealing them only when the object is chosen for supervision. Therefore, the active learning setup does not involve adding new examples to the training set but rather unveiling concepts in the existing ones.

The model was initialized by providing supervision on 10 red-squares, that was sufficient to allow it to achieve optimal accuracy (by learning a reasoning shortcut). Notice that without any initial concept-level supervision, the model was incapable of achieving decent accuracy results because of the complexity of the knowledge.

At each step of the active learning setup, both DPL and bears compute the Shannon entropy on an object, defined as:

$$H(\mathbf{s}, \mathbf{c}|x) = -\sum_{i,j} p_\theta(s_i, c_j|x) \log p_\theta(s_i, c_j|x) \tag{30}$$

where $p_\theta(s_i, c_j|x)$ is the probability of shape $s_i$ and color $c_j$ for object $x$. Plain DPL computes the probability of a certain configuration of concepts for an object $x$ as:

$$p_\theta(s_i, c_j|x) = p_\theta(s_i|x)p_\theta(c_j|x) \tag{31}$$

while `bears` computes it as:

$$p_{\boldsymbol{\theta}}(s_i, c_j|x) = \frac{1}{|\boldsymbol{\theta}|} \sum_{\theta' \in \boldsymbol{\theta}} p_{\theta'}(s_i|x) p_{\theta'}(c_j|x) \tag{32}$$

Where $\boldsymbol{\theta}$ is the learned ensemble. The top 10 elements with largest entropy are then selected to acquire concept-level supervision. The baseline method `DPL + random` ignores concept uncertainty altogether and simply chooses 10 random elements from the training set.

## A.8 RUNTIME COMPARISON

Table 9: Wall-clock time for a single batch in `MNIST-Half`

| METHOD | TRAIN - BATCH | INFERENCE - BATCH | PRE-PROCESS |
|---|---|---|---|
| DPL | 0.011 | 0.001 | |
| DPL + MCDO | | 0.026 | |
| DPL + LA | | 0.019 | 32.249 |
| DPL + PCBM | 0.043 | 0.043 | |
| DPL + DE | 0.017 | 0.027 | |
| DPL + bears | 0.073 | 0.018 | |
| SL | 0.010 | 0.001 | |
| SL + MCDO | | 0.017 | |
| SL + LA | | 0.013 | 21.990 |
| SL + PCBM | 0.043 | 0.054 | |
| SL + DE | 0.033 | 0.020 | |
| SL + bears | 0.085 | 0.014 | |
| LTN | 0.010 | 0.001 | |
| LTN + MCDO | | 0.018 | |
| LTN + LA | | 0.014 | 26.434 |
| LTN + PCBM | 0.035 | 0.045 | |
| LTN + DE | 0.031 | 0.017 | |
| LTN + bears | 0.072 | 0.011 | |

To estimate the order of magnitude of `bears`, we measured the wall-clock time of a run on `MNIST-Half`. Specifically, we computed the wall-clock time of the model inference on a single batch (all batches have the same dimension, i.e. 64). For both `DE` and `bears`, we evaluated only a single model of the ensemble, namely the last model out of 5. In this way, we isolate the time from the number of ensembles. We do not report the training time for `MCDO` and `LA` as they are applied on pre-trained models. Additionally, we account for the pre-processing time of `LA`, which is needed to compute the Hessian matrix for the Laplace approximation. However, it is important to note that this step is done only once.

As shown in Table 9, the inference time of `bears` is comparable to all the competitors, as long as the ensemble is not too big. In terms of training, although we take more time due to the overhead associated with the retrieval of the $p(C|x)$ for ensemble members and the computation of the loss function, we are comparable with `DE`.

## B THEORETICAL MATERIAL

In this section, we include the proofs and the theoretical material needed for the main text. Before moving to the proofs of the main text claims, we report the statement of Lemma 1, Theorem 2, and Proposition 3 from [3] for ease of comparison. These rely on two assumptions, cf. Section 3:

**A1**. *Invertibility*: Each $\mathbf{x}$ is generated by a unique $\mathbf{g}$, *i.e.*, there exists a function $f : \mathbf{x} \mapsto \mathbf{g}$ such that $p^*(\mathbf{G} \mid \mathbf{X}) = \mathbb{1}\{\mathbf{G} - f(\mathbf{X})\}$.

**A2**. *Determinism*: The knowledge $\mathsf{K}$ is *deterministic*, *i.e.*, there exists a function $\beta_{\mathsf{K}} : \mathbf{g} \mapsto \mathbf{y}$ such that $p^*(\mathbf{Y} \mid \mathbf{G}; \mathsf{K}) = \mathbb{1}\{\mathbf{Y} = \beta_{\mathsf{K}}(\mathbf{g})\}$.

We begin by reporting three useful results from [3] that will be used in our proofs. First of all, we will indicate with $\mathrm{supp}(\mathbf{G})$ the support of the probability distribution given by $p^*(\mathbf{G})$.

**Lemma 4.** *It holds that: (ii) Under A1, there exists a bijection between the deterministic concept distributions $p_\theta(\mathbf{C} \mid \mathbf{X})$ that are constant over the support of $p(\mathbf{X} \mid \mathbf{g})$, for each $\mathbf{g} \in \mathrm{supp}(\mathbf{G})$, and the deterministic distributions of the form $p_\theta(\mathbf{C} \mid \mathbf{G})$.*

**Theorem 5.** *Let $\mathcal{A}$ be the set of mappings $\alpha : \mathbf{g} \mapsto \mathbf{c}$ induced by all possible deterministic distributions $p_\theta(\mathbf{C} \mid \mathbf{G})$, i.e., each $p_\theta(\mathbf{C} \mid \mathbf{G}) = \mathbb{1}\{\mathbf{C} = \alpha(\mathbf{G})\}$ for exactly one $\alpha \in \mathcal{A}$. Under A1 and A2, the number of deterministic optima $p_\theta(\mathbf{C} \mid \mathbf{G})$ is:*

$$\sum_{\alpha \in \mathcal{A}} \mathbb{1}\left\{ \bigwedge_{\mathbf{g} \in \mathrm{supp}(\mathbf{G})} (\beta_\mathsf{K} \circ \alpha)(\mathbf{g}) = \beta_\mathsf{K}(\mathbf{g}) \right\} \tag{33}$$

*In particular, the set of optimal maps $\mathcal{A}^*$ is given by:*

$$\mathcal{A}^* = \left\{ \alpha \in \mathcal{A} : \bigwedge_{\mathbf{g} \in \mathrm{supp}(\mathbf{G})} (\beta_\mathsf{K} \circ \alpha)(\mathbf{g}) = \beta_\mathsf{K}(\mathbf{g}) \right\} \tag{34}$$

**Proposition 6.** *For probabilistic logic approaches (including DPL and SL): (i) All convex combinations of two or more deterministic optima $p_\theta(\mathbf{C} \mid \mathbf{X})$ of the likelihood are also (non-deterministic) optima. However, not all convex combinations can be expressed in DPL and SL. (ii) Under A1 and A2, all optima of the likelihood can be expressed as a convex combination of deterministic optima. (iii) If A2 does not hold, there may exist non-deterministic optima that are not convex combinations of deterministic ones. These may be the only optima.*

## B.1 PROOF OF LEMMA 1

**Lemma.** *Take any input-concept distribution $p(\mathbf{C} \mid \mathbf{X})$ and let $p(\mathbf{C} \mid \mathbf{G})$ be the concept-concept distribution entailed by it. Then there exists (at least one) vector $\boldsymbol{\omega}$ such that $p$ is a convex combination of maps $\alpha \in \mathcal{A}$, that is:*

$$p(\mathbf{C} \mid \mathbf{G}) = \sum_{\alpha \in \mathcal{A}} \omega_\alpha \mathbb{1}\{\mathbf{C} = \alpha(\mathbf{G})\} := p_\omega(\mathbf{C} \mid \mathbf{G})$$

*parameterized by $\boldsymbol{\omega} \geq 0$, $\|\boldsymbol{\omega}\|_1 = 1$. Moreover, under invertibility (A1) and determinism (A2), the set of all maps $\mathcal{A}$ restricts to the set of optimal maps $\mathcal{A}^*$.*

*Proof.* By definition, $p(\mathbf{C} \mid \mathbf{G})$ is given by:

$$p(\mathbf{C} \mid \mathbf{G}) := \mathbb{E}_{p(\mathbf{x}\mid\mathbf{g})} p(\mathbf{C} \mid \mathbf{x}) \tag{35}$$

For each $\mathbf{g} \in \{0,1\}^k$, $p(\mathbf{C} \mid \mathbf{g})$ can be written as a convex combination of the maps $\alpha \in \mathcal{A}$:

$$\begin{aligned} p(\mathbf{C} \mid \mathbf{g}) &= \sum_{\mathbf{c}} p(\mathbf{c} \mid \mathbf{g}) \mathbb{1}\{\mathbf{C} = \mathbf{c}\} \\ &= \sum_{\mathbf{c}} \left[ \sum_{\alpha \in \mathcal{A}} \omega_\alpha \mathbb{1}\{\alpha(\mathbf{g}) = \mathbf{c}\} \right] \mathbb{1}\{\mathbf{C} = \mathbf{c}\} \\ &= \sum_{\alpha \in \mathcal{A}} \omega_\alpha \mathbb{1}\{\mathbf{C} = \alpha(\mathbf{g})\} \end{aligned} \tag{36}$$

where in the last line we swapped the summation and used the condition that $\alpha(\mathbf{g}) = \mathbf{c}$. Altogether, this yields for the single $\mathbf{g}$ that the following must hold:

$$\sum_{\alpha \in \mathcal{A}: \alpha(\mathbf{g})=\mathbf{c}} \omega_\alpha = p(\mathbf{c} \mid \mathbf{g}) \tag{37}$$

Combining all the cases this gives a system of $2^k \cdot 2^k$ equations, one for each $\mathbf{g}$ and for each $\mathbf{c}$, for a total of $(2^k)^{2^k}$ variables $\boldsymbol{\omega}$:

$$\sum_{\alpha \in \mathcal{A}: \alpha(\mathbf{g})=\mathbf{c}} \omega_\alpha = p(\mathbf{c} \mid \mathbf{g}), \quad \forall \mathbf{g} \in \{0,1\}^k, \ \forall \mathbf{c} \in \{0,1\}^k \tag{38}$$

This shows that the linear system can always be solved, proving that $\mathcal{A}$ spans the space of $p(\mathbf{C} \mid \mathbf{G})$.

Next, under A1 and A2 we have that to have an optimal model we only need to consider the optimal elements $\alpha \in \mathcal{A}^*$, where the set $\mathcal{A}^*$ is defined from Theorem 5:

$$\mathcal{A}^* = \left\{ \alpha \in \mathcal{A} : \bigwedge_{\mathbf{g} \in \mathrm{supp}(\mathbf{G})} (\beta_\mathsf{K} \circ \alpha)(\mathbf{g}) = \beta_\mathsf{K}(\mathbf{g}) \right\} \tag{39}$$

We proceed by contradiction. Suppose there exists one $\alpha' \notin \mathcal{A}^*$ such that:

$$p(\mathbf{C} \mid \mathbf{G}) = \omega(\alpha') \mathbb{1}\{\mathbf{C} = \alpha'(\mathbf{g})\} + \sum_{\alpha \in \mathcal{A}^*} \omega_\alpha \mathbb{1}\{\mathbf{C} = \alpha(\mathbf{g})\} \tag{40}$$

is still optimal. Notice that there exists at least one $\mathbf{g}$ such that $\alpha'(\mathbf{g}) \neq \alpha(\mathbf{g})$, $\forall \alpha \in \mathcal{A}^*$. This means that for those values $\mathbf{g}$ we have $(\beta_{\mathsf{K}} \circ \alpha')(\mathbf{g}) \neq \beta_{\mathsf{K}}(\mathbf{g})$. Therefore, the NeSy predictor will result in a suboptimal model, since it does not place the mass on concepts attaining the same label. This proves the contradiction, yielding the claim. $\qquad \square$

## B.2 ENTROPY ON CONCEPT VECTORS AND REASONING SHORTCUTS

Under *invertibility* (**A1**) and *determinism* (**A2**), it is possible to describe entirely the set of optimal maps $\alpha : \mathbf{G} \mapsto \mathbf{C}$ through Theorem 5, which we denote with $\mathcal{A}^*$. Before moving on to prove the main results in the main text, it is useful to introduce here the notion of "equivalence set" of a concept vector $\mathbf{g}$ given the optimal maps $\alpha \in \mathcal{A}^*$:

$$\mathcal{E}(\mathbf{g}; \mathsf{K}) = \{\alpha(\mathbf{g}) : \ \forall \alpha \in \mathcal{A}^*\} \tag{41}$$

that contains all the concepts $\mathbf{c} \in \{0,1\}^k$ that are predicted by the maps $\alpha \in \mathcal{A}^*$. With this, we can formally define when a ground-truth concept $\mathbf{g}$ is mispredicted by RSs:

**Definition 1.** *We say that a concept vector $\mathbf{g} \in \{0,1\}^k$ is "mispredicted" by RSs when $|\mathcal{E}(\mathbf{g}; \mathsf{K})| > 1$, i.e., there exist at least two different $\alpha_i, \alpha_j \in \mathcal{A}^*$, such that $\alpha_i(\mathbf{g}) \neq \alpha_j(\mathbf{g})$. Conversely, a concept vector is "correctly predicted" if $\alpha(\mathbf{g}) = \mathbf{g}$, $\forall \alpha \in \mathcal{A}^*$.*

Following, we can use the decomposition in terms of the map $\alpha$'s to inspect what combinations with optimal weights $\boldsymbol{\omega}^* = (\omega_1, \ldots, \omega_N)$, where $N = |\mathcal{A}^*|$ and $\|\boldsymbol{\omega}^*\|_1 = 1$, give high entropy on single concepts $\mathbf{g} \in \{0,1\}^k$:

**Proposition 7.** *Suppose that $p_{\boldsymbol{\omega}}(\mathbf{C} \mid \mathbf{G})$ admits a decomposition as a weighted sum of at least two distinct $\alpha \in \mathcal{A}$, with weights $\boldsymbol{\omega}$. Then,*

(i) *for any $\mathbf{g} \in \{0,1\}^k$ it holds that $H(p_{\boldsymbol{\omega}}(\mathbf{C} \mid \mathbf{g})) = 0$, when $\forall \alpha_i, \alpha_j \in \mathcal{A}$ such that $\omega_{\alpha_i} > 0$ and $\omega_{\alpha_j} > 0$, it holds $\alpha_i(\mathbf{g}) = \alpha_j(\mathbf{g})$.*

*Assuming that **A1** and **A2** hold:*

(ii) *If a concept $\mathbf{g} \in \{0,1\}^k$ is not mispredicted by RSs, then all combinations $\boldsymbol{\omega}^*$ of $\alpha \in \mathcal{A}^*$ will give zero entropy $H(p_{\boldsymbol{\omega}^*}(\mathbf{C} \mid \mathbf{g}))$*

(iii) *Vice versa, if $\mathbf{g}$ is mispredicted by RSs, there is always at least one combination $\boldsymbol{\omega}^*$ such that the entropy $H(p_{\boldsymbol{\omega}^*}(\mathbf{C} \mid \mathbf{g}))$ attains a maximal value of:*
$$H(p_{\boldsymbol{\omega}^*}(\mathbf{C} \mid \mathbf{g})) = \log |\mathcal{E}(\mathbf{g}; \mathsf{K})| \tag{42}$$

*Proof.* (*i*) We start by considering a $p_{\boldsymbol{\omega}}(\mathbf{C} \mid \mathbf{G})$ given by a fixed convex combination of maps $\alpha \in \mathcal{A}$, with a vector $\boldsymbol{\omega}$. We proceed to show that for any $\mathbf{g} \in \{0,1\}^k$, the entropy is zero holds *if and only if* $\forall \alpha_i, \alpha_j \in \mathcal{A}$ with $\omega(\alpha_i) > 0$ and $\omega(\alpha_j) > 0$, we have that $\alpha_i(\mathbf{g}) = \alpha_j(\mathbf{g})$.

We consider a vanishing conditional entropy $H(p(\mathbf{C} \mid \mathbf{g}))$ that is given only when $p(\mathbf{C} \mid \mathbf{g}) = \mathbb{1}\{\mathbf{C} = \mathbf{c}\}$, for $\mathbf{c} \in \{0,1\}^k$. This occurs only if (1) $p(\mathbf{C} \mid \mathbf{g}) = \mathbb{1}\{\mathbf{C} = \alpha(\mathbf{g})\}$, for $\alpha \in \mathcal{A}$, or if (2) $p(\mathbf{C} \mid \mathbf{g}) = \sum_{\alpha \in \mathcal{A}} \omega_\alpha \mathbb{1}\{\mathbf{C} = \alpha(\mathbf{g})\}$, with $\omega_\alpha > 0$ only if $\alpha(\mathbf{g})$ is the same. Since we are considering probabilities $p(\mathbf{C} \mid \mathbf{g})$ with at least two $\alpha$'s, only (2) holds, proving the result.

(*ii*) Next, under **A1** and **A2**, we consider the case where we have optimal maps $\alpha \in \mathcal{A}^*$. For those $\mathbf{g}$'s that are *correctly predicted* even by RSs, by definition $\mathcal{E}(\mathbf{g}; \mathcal{A}^*) = 1$, and in particular $\alpha(\mathbf{g}) = \mathbf{g}$ for all $\alpha \in \mathcal{A}^*$. This means that whatever combination of weights $\boldsymbol{\omega}^*$ is chosen, there will be only one element for $p(\mathbf{C} \mid \mathbf{g})$ with all the probability mass. Therefore:

$$p_{\boldsymbol{\omega}^*}(\mathbf{C} \mid \mathbf{G}) = \sum_{\alpha \in \mathcal{A}^*} \omega_\alpha^* \mathbb{1}\{\mathbf{C} = \alpha(\mathbf{g})\} = \mathbb{1}\{\mathbf{C} = \mathbf{g}\} \sum_{\alpha \in \mathcal{A}^*} \omega_\alpha^* \tag{43}$$

that leads to a vanishing entropy.

(*iii*) For any optimal solution, it holds that:

$$\text{supp}(p(\mathbf{C} \mid \mathbf{g})) \subseteq \mathcal{E}(\mathbf{g}; \mathcal{A}^*), \quad \forall \mathbf{g} \in \{0,1\}^k \tag{44}$$

since all concept vectors having non-zero mass in $p(\mathbf{C} \mid \mathbf{g})$ must be optimal. We now consider a concept vector $\mathbf{g}$ that is affected by RSs, in that there exists $\alpha_i, \alpha_j \in \mathcal{A}^*$ such that $\alpha_i(\mathbf{g}) \neq \alpha_j(\mathbf{g})$. We then rewrite it as follows:

$$
\begin{aligned}
p_{\boldsymbol{\omega}^*}(\mathbf{C} \mid \mathbf{g}) &= \sum_{\alpha \in \mathcal{A}^*} \omega_\alpha^* \mathbb{1}\{\mathbf{C} = \alpha(\mathbf{g})\} \\
&= \sum_{\mathbf{c} \in \mathcal{E}(\mathbf{g}; \mathcal{A}^*)} \Big[ \sum_{\alpha \in \mathcal{A}^*: \alpha(\mathbf{g}) = \mathbf{c}} \omega_\alpha^* \Big] \mathbb{1}\{\mathbf{C} = \mathbf{c}\} \\
&= \sum_{\mathbf{c} \in \mathcal{E}(\mathbf{g}; \mathcal{A}^*)} \lambda_{\mathbf{c}} \mathbb{1}\{\mathbf{C} = \mathbf{c}\}
\end{aligned}
\tag{45}
$$

where we denoted $\lambda_{\mathbf{c}} = \sum_{\alpha \in \mathcal{A}^*: \alpha(\mathbf{g}) = \mathbf{c}} \omega^*(\alpha)$ the weight associated to $\mathbb{1}\{\mathbf{C} = \mathbf{c}\}$. When plugging this into the entropy we have that:

$$
\begin{aligned}
H(p(\mathbf{C} \mid \mathbf{g})) &= - \sum_{\mathbf{c} \in \{0,1\}^k} p(\mathbf{c} \mid \mathbf{g}) \log p(\mathbf{c} \mid \mathbf{g}) \\
&= - \sum_{\mathbf{c} \in \{0,1\}^k} \sum_{\mathbf{c}' \in \mathcal{E}(\mathbf{g}; \mathcal{A}^*)} \lambda_{\mathbf{c}'} \mathbb{1}\{\mathbf{c}' = \mathbf{c}\} \log \sum_{\mathbf{c}' \in \mathcal{E}(\mathbf{g}; \mathcal{A}^*)} \lambda_{\mathbf{c}'} \mathbb{1}\{\mathbf{c}' = \mathbf{c}\} \\
&= - \sum_{\mathbf{c} \in \mathcal{E}(\mathbf{g}; \mathcal{A}^*)} \lambda_{\mathbf{c}} \log \lambda_{\mathbf{c}} \\
&\leq - \sum_{\mathbf{c} \in \mathcal{E}(\mathbf{g}; \mathcal{A}^*)} \frac{1}{|\mathcal{E}(\mathbf{g}; \mathcal{A}^*)|} \log \frac{1}{|\mathcal{E}(\mathbf{g}; \mathcal{A}^*)|} \\
&= \log |\mathcal{E}(\mathbf{g}; \mathcal{A}^*)|
\end{aligned}
\tag{46}
$$

where the equality holds if and only if $\lambda_{\mathbf{c}} = \frac{1}{|\mathcal{E}(\mathbf{g}; \mathcal{A}^*)|}$ for all $\mathbf{c}$ in $p(\mathbf{c} \mid \mathbf{g})$. We can therefore choose $\omega^*$ such that:

$$\sum_{\alpha \in \mathcal{A}^*: \alpha(\mathbf{g}) = \mathbf{c}} \omega^*(\alpha) = |\mathcal{E}(\mathbf{g}; \mathcal{A}^*)|^{-1}, \quad \forall \mathbf{c} \in \mathcal{E}(\mathbf{g}; \mathcal{A}^*) \tag{47}$$

which fixes $|\mathcal{E}(\mathbf{g}; \mathcal{A}^*)|$ equations for at least $|\mathcal{E}(\mathbf{g}; \mathcal{A}^*)|$ variables $\omega^*$. In fact, the number of maps $\alpha \in \mathcal{A}^*$ is equal to $|\mathcal{E}(\mathbf{g}; \mathcal{A}^*)|$ only when all maps $\alpha_i(\mathbf{g}) \neq \alpha_j(\mathbf{g}), \forall \alpha_i, \alpha_j \in \mathcal{A}^*$, *i.e.*, there are not two different maps that predict the same concept for $\mathbf{g}$. This shows that by choosing the coefficients $\omega_\alpha$ correctly, it is possible to obtain a maximally entropic distribution $p(\mathbf{C} \mid \mathbf{g})$. This concludes the proof. $\qquad\square$

Point (*ii*) of Proposition 7 essentially captures the intuition that concept vectors that are "correctly predicted" even by RSs will not contribute to increasing the entropy of the distribution $p(\mathbf{C} \mid \mathbf{G})$. Conversely, when a concept is "mispredicted" by RSs, there is always a combination attaining maximal entropy from point (*iii*). Achieving maximal entropy for one ground-truth concept, however, is not enough to guarantee the others will also display maximal entropy. This can happen because a combination $\boldsymbol{\omega}^*$ may increase the entropy of one $\mathbf{g}_i$ while decreasing that of another $\mathbf{g}_j$.

## B.3 PROOF OF PROPOSITION 2

Before proceeding, it is useful to pin down what we mean precisely with the set of parameters. Based on the generative process with $p^*(\mathbf{Y} \mid \mathbf{G}; \mathsf{K})$, we define as "optimal" those parameters $\theta$ that meet the following criterion:

$$p_\theta(\mathbf{Y} \mid \mathbf{G}; \mathsf{K}) := \int p_\theta(\mathbf{Y} \mid \mathbf{x}; \mathsf{K}) p(\mathbf{x} \mid \mathbf{G}) \mathrm{d}\mathbf{x} = p^*(\mathbf{Y} \mid \mathbf{G}; \mathsf{K}) \tag{48}$$

and denote the whole set with $\Theta^*$.

**Proposition.** *Consider only optimal parameters $\theta \in \Theta^*$ for $p_\theta(\mathbf{C} \mid \mathbf{G})$. Assuming that $p_\theta$ is expressive enough to capture every possible combination $p_{\boldsymbol{\omega}}$, i.e., for each $\boldsymbol{\omega}$ there exists $\theta$ s.t. $p_\theta(\mathbf{C} \mid \mathbf{G}) = p_{\boldsymbol{\omega}}(\mathbf{C} \mid \mathbf{G})$, under invertibility (A1) and determinism (A2), it holds that:*

$$\max_{\theta \in \Theta^*} H(p_\theta(\mathbf{C} \mid \mathbf{G})) = \max_{\boldsymbol{\omega}^*} H(p_{\boldsymbol{\omega}^*}(\mathbf{C} \mid \mathbf{G}))$$

*Proof.* We start from the fact that by [Lemma 1](#) we can always express $p_\theta(\mathbf{C} \mid \mathbf{G})$ as a convex combination of maps $\alpha \in \mathcal{A}^*$ for some weights $\boldsymbol{\omega}^*(\theta)$. Vice versa, since $p_\theta$ is flexible enough to capture any combination $\boldsymbol{\omega}^*$ for $p_{\boldsymbol{\omega}^*}(\mathbf{C} \mid \mathbf{G})$, there will exists some weights $\theta(\boldsymbol{\omega}^*)$ associated to any vector $\boldsymbol{\omega}^*$. Notice that, in general, neither $\boldsymbol{\omega}^*(\theta)$ nor $\theta(\boldsymbol{\omega}^*)$ are unique. This, nonetheless, allows us to convert a problem formulated in terms $\theta \in \Theta^*$ to one in terms of $\boldsymbol{\omega}^*$:

$$
\begin{aligned}
\max_{\theta \in \Theta^*} H(p_\theta(\mathbf{C} \mid \mathbf{G})) &= \max_{\theta \in \Theta^*} - \sum_{\mathbf{g} \in \{0,1\}^{2k}} p^*(\mathbf{g}) \sum_{\mathbf{c} \in \{0,1\}^k} p_\theta(\mathbf{c} \mid \mathbf{g}) \log p_\theta(\mathbf{c} \mid \mathbf{g}) \\
&= \max_{\theta \in \Theta^*} - \sum_{\mathbf{g} \in \{0,1\}^{2k}} p^*(\mathbf{g}) \sum_{\mathbf{c} \in \{0,1\}^k} \left[ \sum_{\alpha \in \mathcal{A}^*} \omega_\alpha^*(\theta) \mathbb{1}\{\mathbf{c} = \alpha(\mathbf{g})\} \log \sum_{\alpha' \in \mathcal{A}^*} \omega_{\alpha'}^*(\theta) \mathbb{1}\{\mathbf{c} = \alpha'(\mathbf{g})\} \right] \\
&= \max_{\boldsymbol{\omega}^*, \, ||\boldsymbol{\omega}^*||_1 = 1} - \sum_{\mathbf{g} \in \{0,1\}^{2k}} p^*(\mathbf{g}) \sum_{\mathbf{c} \in \{0,1\}^k} \left[ \sum_{\alpha \in \mathcal{A}^*} \omega^*(\alpha) \mathbb{1}\{\mathbf{c} = \alpha(\mathbf{g})\} \log \sum_{\alpha \in \mathcal{A}^*} \omega^*(\alpha) \mathbb{1}\{\mathbf{c} = \alpha(\mathbf{g})\} \right] \\
&= \max_{\boldsymbol{\omega}^*, \, ||\boldsymbol{\omega}^*||_1 = 1} - \sum_{\mathbf{g} \in \{0,1\}^{2k}} p^*(\mathbf{g}) \sum_{\mathbf{c} \in \{0,1\}^k} p_{\boldsymbol{\omega}^*}(\mathbf{c} \mid \mathbf{g}) \log p_{\boldsymbol{\omega}^*}(\mathbf{c} \mid \mathbf{g}) \\
&= \max_{\boldsymbol{\omega}^*, \, ||\boldsymbol{\omega}^*||_1 = 1} H(p_{\boldsymbol{\omega}^*}(\mathbf{C} \mid \mathbf{G}))
\end{aligned}
\tag{49}
$$

where in the third line we converted the maximization problem on the parameters $\theta \in \Theta^*$ to the weights $\boldsymbol{\omega}^*$. This concludes the proof. $\qquad \square$

### B.4  PROOF OF PROPOSITION [3](#)

**Proposition.** *Let $p(\mathbf{C} \mid \mathbf{X})$ be given by a convex combination of models $p_{\theta_i}(\mathbf{C} \mid \mathbf{X})$, for $i \in [K]$, where $K$ denotes the total number of components of $\boldsymbol{\theta} = \{\theta_i\}$, and each $\theta_i \in \Theta^*$. Let also $\boldsymbol{\lambda} = \{\lambda_i\}$ contain all the weights $\lambda_i$ associated to each component $\theta_i$. Under invertibility (A1) and determinism (A2), there exists $K \leq |\mathcal{A}^*|$ such that maximizing the entropy of $p_{\boldsymbol{\omega}^*}(\mathbf{C} \mid \mathbf{G})$ can be solved by maximizing $H(p_{\boldsymbol{\theta}}(\mathbf{C} \mid \mathbf{X}))$ on $\boldsymbol{\theta}$ and $\boldsymbol{\lambda}$, that is:*

$$
\max_{\boldsymbol{\theta}, \boldsymbol{\lambda}} H\Big( \sum_{i=1}^{K} \lambda_i p_{\theta_i}(\mathbf{C} \mid \mathbf{X}) \Big) = \max_{\boldsymbol{\omega}^*} H(p_{\boldsymbol{\omega}^*}(\mathbf{C} \mid \mathbf{G}))
$$

*Furthermore, we can write the maximization of $H(p_{\boldsymbol{\theta}}(\mathbf{C} \mid \mathbf{X}))$ as:*

$$
\max_{\boldsymbol{\theta}, \boldsymbol{\lambda}} \int p(\mathbf{x}) \sum_{i=1}^{K} \lambda_i [\mathsf{KL}(p_{\theta_i}(\mathbf{c} \mid \mathbf{x}) \, || \, \sum_{j=1}^{K} \lambda_j p_{\theta_j}(\mathbf{c} \mid \mathbf{x})) + H(p_{\theta_i}(\mathbf{C} \mid \mathbf{x}))] \mathrm{d}\mathbf{x}
$$

*where* $\mathsf{KL}$ *denotes the Kullback-Lieber divergence.*

*Proof.* We start with $p(\mathbf{C} \mid \mathbf{X}) = \sum_i \lambda_i p_{\theta_i}(\mathbf{C} \mid \mathbf{X})$ given by a convex combination of optimal models with parameters $\theta_i$, each entailing a deterministic distribution $p_{\theta_i}(\mathbf{C} \mid \mathbf{G}) = \mathbb{1}\{\mathbf{C} = \alpha_i(\mathbf{G})\}$.

Recall that, by invertibility (A1), there exists $f : \mathbf{x} \mapsto \mathbf{g}$, entailing the inverse of $p^*(\mathbf{G} \mid \mathbf{X})$. We know that by [Lemma 4](#) (*ii*), if $p_{\theta_i}$ entails a deterministic distributions $p_{\theta_i}(\mathbf{C} \mid \mathbf{G}) = \mathbb{1}\{\mathbf{C} = \alpha_i(\mathbf{G})\}$, then it is in one-to-one correspondence with $p_{\theta_i}(\mathbf{C} \mid \mathbf{X})$. Formally, the latter is:

$$
p_{\theta_i}(\mathbf{C} \mid \mathbf{x}') = \mathbb{1}\{\mathbf{C} = \alpha_i(\mathbf{g})\}, \quad \forall \mathbf{x}' \in \mathsf{supp}(p^*(\mathbf{X} \mid \mathbf{g}), \text{ where } \mathbf{g} = f(\mathbf{x})
\tag{50}
$$

Now, from the above equation, we can rewrite $H(p_{\boldsymbol{\theta}}(\mathbf{C} \mid \mathbf{X}))$ as follows:

$$
\begin{aligned}
H(p_{\boldsymbol{\theta}}(\mathbf{C} \mid \mathbf{X})) &= -\mathbb{E}_{p^*(\mathbf{x})}\Big[ \sum_{\mathbf{c} \in \{0,1\}^k} p_{\boldsymbol{\theta}}(\mathbf{c} \mid \mathbf{x}) \log p_{\boldsymbol{\theta}}(\mathbf{c} \mid \mathbf{x}) \Big] \\
&= - \sum_{\mathbf{g} \in \{0,1\}^k} p^*(\mathbf{g}) \mathbb{E}_{p^*(\mathbf{x} \mid \mathbf{g})} \Big[ \sum_{\mathbf{c} \in \{0,1\}^k} \sum_{i=1}^K \lambda_i p_{\theta_i}(\mathbf{c} \mid \mathbf{x}) \log \sum_{j=1}^K \lambda_j p_{\theta_j}(\mathbf{c} \mid \mathbf{x}) \Big] \\
&= - \sum_{\mathbf{g} \in \{0,1\}^k} p^*(\mathbf{g}) \sum_{\mathbf{c} \in \{0,1\}^k} \mathbb{E}_{p^*(\mathbf{x} \mid \mathbf{g})} \Big[ \sum_{i=1}^K \lambda_i p_{\theta_i}(\mathbf{c} \mid \mathbf{x}) \log \sum_{j=1}^K \lambda_j p_{\theta_j}(\mathbf{c} \mid \mathbf{x}) \Big] \\
&= - \sum_{\mathbf{g} \in \{0,1\}^k} p^*(\mathbf{g}) \sum_{\mathbf{c} \in \{0,1\}^k} \mathbb{E}_{p^*(\mathbf{x} \mid \mathbf{g})} \Big[ \sum_{i=1}^K \lambda_i \mathbb{1}\{\mathbf{c} = \alpha_i(\mathbf{g})\} \log \sum_{j=1}^K \lambda_j \mathbb{1}\{\mathbf{c} = \alpha_j(\mathbf{g})\} \Big] \\
&= - \sum_{\mathbf{g} \in \{0,1\}^k} p^*(\mathbf{g}) \sum_{\mathbf{c} \in \{0,1\}^k} \sum_{i=1}^K \lambda_i \mathbb{1}\{\mathbf{c} = \alpha_i(\mathbf{g})\} \log \sum_{j=1}^K \lambda_j \mathbb{1}\{\mathbf{c} = \alpha_j(\mathbf{g})\} \\
&= - \sum_{\mathbf{g} \in \{0,1\}^k} p^*(\mathbf{g}) \sum_{\mathbf{c} \in \{0,1\}^k} p_{\boldsymbol{\theta}}(\mathbf{c} \mid \mathbf{g}) \log p_{\boldsymbol{\theta}}(\mathbf{c} \mid \mathbf{g}) \\
&= H(p_{\boldsymbol{\theta}}(\mathbf{C} \mid \mathbf{G}))
\end{aligned}
\tag{51}
$$

where the second line follows from the fact that the expectation on the input variables can be written as $\mathbb{E}_{p^*(\mathbf{g})}[p^*(\mathbf{x} \mid \mathbf{g})]$, and $p_{\boldsymbol{\theta}}(\mathbf{C} \mid \mathbf{G})$ is the distribution with convex weights $\boldsymbol{\lambda}$, where each $\lambda_i$ is associated to the reasoning shortcut $\alpha_i$, entailed by $\theta_i$. This means that maximizing $H(p_{\boldsymbol{\theta}}(\mathbf{C} \mid \mathbf{X}))$ directly maximizes $H(p_{\boldsymbol{\theta}}(\mathbf{C} \mid \mathbf{G}))$.

Next, suppose that $\boldsymbol{\theta}$ is fixed and contains a number $K = |\mathcal{A}^*|$ of members, such that each deterministic RS $\alpha \in \mathcal{A}^*$ is captured by exactly one member $\theta_i \in \boldsymbol{\theta}$. This means that each $\theta_i$ captures $p_{\theta_i}(\mathbf{C} \mid \mathbf{G}) = \mathbb{1}\{\mathbf{C} = \alpha_i(\mathbf{G})\}$, and it holds that if $\theta_i \neq \theta_j$, then $\alpha_i(\mathbf{g}) \neq \alpha_j(\mathbf{g})$ for at least one $\mathbf{g} \in \{0,1\}^k$.

We prove that maximizing $\boldsymbol{\lambda}$ when $\boldsymbol{\theta}$ is fixed and contains all possible deterministic RSs amounts to maximizing the combination of RSs. The proof follows a similar derivation to [Proposition 2](#):

$$
\begin{aligned}
\max_{\boldsymbol{\lambda}, ||\boldsymbol{\lambda}||_1 = 1} H(p_{\boldsymbol{\theta}}(\mathbf{C} \mid \mathbf{G})) &= \max_{\boldsymbol{\lambda}, ||\boldsymbol{\lambda}||_1 = 1} - \sum_{\mathbf{g} \in \{0,1\}^k} p^*(\mathbf{g}) \sum_{\mathbf{c} \in \{0,1\}^k} p_{\boldsymbol{\theta}}(\mathbf{c} \mid \mathbf{g}) \log p_{\boldsymbol{\theta}}(\mathbf{c} \mid \mathbf{g}) \\
&= \max_{\boldsymbol{\lambda}, ||\boldsymbol{\lambda}||_1 = 1} - \sum_{\mathbf{g} \in \{0,1\}^k} p^*(\mathbf{g}) \sum_{\mathbf{c} \in \{0,1\}^k} \Big[ \sum_{i=1}^K \lambda_i \mathbb{1}\{\mathbf{c} = \alpha_i(\mathbf{g})\} \log \sum_{j=1}^K \lambda_j \mathbb{1}\{\mathbf{c} = \alpha_j(\mathbf{g})\} \Big] \\
&= \max_{\boldsymbol{\omega}^*, ||\boldsymbol{\omega}^*||_1 = 1} - \sum_{\mathbf{g} \in \{0,1\}^k} p^*(\mathbf{g}) \sum_{\mathbf{c} \in \{0,1\}^k} \Big[ \sum_{\alpha \in \mathcal{A}^*} \omega_\alpha^* \mathbb{1}\{\mathbf{c} = \alpha(\mathbf{g})\} \log \sum_{\alpha' \in \mathcal{A}^*} \omega_{\alpha'}^* \mathbb{1}\{\mathbf{c} = \alpha'(\mathbf{g})\} \Big] \\
&= \max_{\boldsymbol{\omega}^*, ||\boldsymbol{\omega}^*||_1 = 1} - \sum_{\mathbf{g} \in \{0,1\}^k} p^*(\mathbf{g}) \sum_{\mathbf{c} \in \{0,1\}^k} p_{\boldsymbol{\omega}^*}(\mathbf{c} \mid \mathbf{g}) \log p_{\boldsymbol{\omega}^*}(\mathbf{c} \mid \mathbf{g}) \\
&= \max_{\boldsymbol{\omega}^*, ||\boldsymbol{\omega}^*||_1 = 1} H(p_{\boldsymbol{\omega}^*}(\mathbf{C} \mid \mathbf{G}))
\end{aligned}
\tag{52}
$$

where in the third line we substituted $\lambda_i$ with $\omega_\alpha^*$ and the summation over the ordered components with the summation over $\alpha \in \mathcal{A}^*$. Notice that this also means that an ensemble containing all different deterministic RSs with parameters $\theta_i$ can express arbitrary combinations of them via $\boldsymbol{\lambda}$.

Now, consider the case where a few elements of $\mathcal{A}^*$ contribute to achieving maximum entropy for $p_{\boldsymbol{\omega}^*}(\mathbf{C} \mid \mathbf{G})$. Therefore, there exists at least one $\omega_{\alpha'}^* = 0$, while the remaining lead to the maximum entropy for $p_{\boldsymbol{\omega}^*}(\mathbf{C} \mid \mathbf{G})$. It holds that, similarly, the maximum of $H(p_{\boldsymbol{\theta}}(\mathbf{C} \mid \mathbf{G}))$ can be obtained by considering a smaller number of components $\boldsymbol{\theta}$ since the weight associated with a specific $\theta_j$ capturing $\alpha'$ must be 0. This also means that the ensemble dimension $K$ can be strictly smaller than $|\mathcal{A}^*|$, while still achieving maximal entropy.

We now maximize the entropy on $\boldsymbol{\theta}$ and $\boldsymbol{\lambda}$ together:

$$
\max_{\boldsymbol{\theta}, \boldsymbol{\lambda}} H(p_{\boldsymbol{\theta}}(\mathbf{C} \mid \mathbf{G}))
\tag{53}
$$

Since the number of components $K$ is upper-bounded by $|\mathcal{A}^*|$, we can always find a solution by getting all different $\theta_i$, each capturing different deterministic distributions $\alpha_i$. On the other hand, when a fewer number of $\alpha$'s are required, it suffices to find those $K$ components $\theta_i$ that are combined with a non-zero weight $\lambda_i$. In this case, $K < |\mathcal{A}^*|$. This means, altogether, that:

$$\max_{\boldsymbol{\theta},\boldsymbol{\lambda}} H(p_{\boldsymbol{\theta}}(\mathbf{C} \mid \mathbf{X})) = \max_{\boldsymbol{\theta},\boldsymbol{\lambda}} H(p_{\boldsymbol{\theta}}(\mathbf{C} \mid \mathbf{G})) = \max_{\boldsymbol{\omega}} H(p_{\boldsymbol{\omega}}(\mathbf{C} \mid \mathbf{G})) \tag{54}$$

proving our first point.

We proceed by analyzing the conditional entropy $H(\mathbf{C} \mid \mathbf{X})$, which can be written as:

$$
\begin{aligned}
H(\mathbf{C} \mid \mathbf{X}) &= -\int p(\mathbf{x}) \sum_{\mathbf{c}\in\{0,1\}^k} p(\mathbf{c} \mid \mathbf{x}) \log p(\mathbf{c} \mid \mathbf{x}) \mathrm{d}\mathbf{x} \\
&= -\int p(\mathbf{x}) \sum_{\mathbf{c}\in\{0,1\}^k} \sum_i \lambda_i p_{\theta_i}(\mathbf{c} \mid \mathbf{x}) \log \sum_j \lambda_j p_{\theta_j}(\mathbf{c} \mid \mathbf{x}) \mathrm{d}\mathbf{x} \\
&= \int p(\mathbf{x}) \sum_{\mathbf{c}\in\{0,1\}^k} \sum_i \lambda_i p_{\theta_i}(\mathbf{c} \mid \mathbf{x}) \left[\log \frac{p_{\theta_i}(\mathbf{c} \mid \mathbf{x})}{\sum_j \lambda_j p_{\theta_j}(\mathbf{c} \mid \mathbf{x})} - \log p_{\theta_i}(\mathbf{c} \mid \mathbf{x})\right] \mathrm{d}\mathbf{x} \\
&= \int p(\mathbf{x}) \sum_i \lambda_i \big[\mathsf{KL}(p_{\theta_i}(\mathbf{c} \mid \mathbf{x}) \mid\mid \sum_j \lambda_j p_{\theta_j}(\mathbf{c} \mid \mathbf{x})) + H(p_{\theta_i}(\mathbf{c} \mid \mathbf{x}))\big] \mathrm{d}\mathbf{x}
\end{aligned}
\tag{55}
$$

where in the third line we multiplied and divided for the members of the ensemble $p_{\theta_i}(\mathbf{c} \mid \mathbf{x})$, and in the last line we grouped the expressions of the $\mathsf{KL}$ divergence and of the conditional entropy. Therefore, for the maximization on $\boldsymbol{\theta}$ and $\boldsymbol{\lambda}$:

$$\max_{\boldsymbol{\lambda},\boldsymbol{\theta}} H(\mathbf{C} \mid \mathbf{X}) = \max_{\boldsymbol{\lambda},\boldsymbol{\theta}} \int p(\mathbf{x}) \sum_i \lambda_i \big[\mathsf{KL}(p_{\theta_i}(\mathbf{c} \mid \mathbf{x}) \mid\mid \sum_j \lambda_j p_{\theta_j}(\mathbf{c} \mid \mathbf{x})) + H(p_{\theta_i}(\mathbf{C} \mid \mathbf{X}))\big] \mathrm{d}\mathbf{x} \tag{56}$$

as claimed. This concludes the proof. $\qquad\square$

# C ADDITIONAL RESULTS

## C.1 MNIST-ADDITION

We report here additional results for the experiments shown in Section 4. Along with $ECE_Y$ and $ECE_C$, we show also the performances of bears compared to other competitors in terms of the label accuracy ($Acc_Y$) and concept accuracy ($Acc_C$), both in-distribution and out-of-distribution.

Table 10: Complete evaluation on MNIST-Half. The values on $Acc_Y$ *in-distribution* shows that bears and all competitors achieve optimal predictions on labels. The values of $Acc_C$ *in-distribution*, on the other hand, show that all methods pick up a RS. This holds for DPL, SL, and LTN. The pattern completely change out-of-distribution, where all methods struggle in terms of label accuracy $Acc_{Yood}$.

| METHOD | $Acc_Y$ | $Acc_C$ | $ECE_Y$ | $ECE_C$ | $Acc_{Yood}$ | $Acc_{Cood}$ | $ECE_{Yood}$ | $ECE_{Cood}$ |
|---|---|---|---|---|---|---|---|---|
| | | | | MNIST-Half | | | | |
| DPL | $0.98 \pm 0.01$ | $0.43 \pm 0.01$ | $0.02 \pm 0.01$ | $0.69 \pm 0.01$ | $0.06 \pm 0.01$ | $0.39 \pm 0.01$ | $0.92 \pm 0.01$ | $0.87 \pm 0.01$ |
| DPL + MCDO | $0.98 \pm 0.01$ | $0.43 \pm 0.01$ | $0.02 \pm 0.01$ | $0.69 \pm 0.01$ | $0.06 \pm 0.01$ | $0.39 \pm 0.01$ | $0.91 \pm 0.01$ | $0.86 \pm 0.01$ |
| DPL + LA | $0.98 \pm 0.01$ | $0.43 \pm 0.01$ | $0.06 \pm 0.01$ | $0.65 \pm 0.01$ | $0.06 \pm 0.01$ | $0.39 \pm 0.01$ | $0.87 \pm 0.01$ | $0.82 \pm 0.01$ |
| DPL + PCBM | $0.98 \pm 0.01$ | $0.43 \pm 0.01$ | $0.07 \pm 0.08$ | $0.64 \pm 0.08$ | $0.06 \pm 0.01$ | $0.39 \pm 0.01$ | $0.86 \pm 0.08$ | $0.80 \pm 0.08$ |
| DPL + DE | $0.99 \pm 0.01$ | $0.43 \pm 0.01$ | $0.01 \pm 0.01$ | $0.64 \pm 0.01$ | $0.06 \pm 0.01$ | $0.39 \pm 0.01$ | $0.83 \pm 0.13$ | $0.77 \pm 0.13$ |
| DPL + bears | $0.99 \pm 0.01$ | $0.43 \pm 0.01$ | $0.09 \pm 0.02$ | $0.37 \pm 0.01$ | $0.06 \pm 0.01$ | $0.39 \pm 0.01$ | $0.39 \pm 0.03$ | $0.38 \pm 0.02$ |
| SL | $0.99 \pm 0.01$ | $0.43 \pm 0.01$ | $0.01 \pm 0.01$ | $0.71 \pm 0.01$ | $0.01 \pm 0.01$ | $0.39 \pm 0.01$ | $0.95 \pm 0.01$ | $0.88 \pm 0.01$ |
| SL + MCDO | $0.99 \pm 0.01$ | $0.43 \pm 0.01$ | $0.01 \pm 0.01$ | $0.70 \pm 0.01$ | $0.01 \pm 0.01$ | $0.39 \pm 0.01$ | $0.92 \pm 0.01$ | $0.88 \pm 0.01$ |
| SL + LA | $0.98 \pm 0.01$ | $0.43 \pm 0.01$ | $0.06 \pm 0.01$ | $0.59 \pm 0.02$ | $0.01 \pm 0.01$ | $0.39 \pm 0.01$ | $0.75 \pm 0.01$ | $0.75 \pm 0.02$ |
| SL + PCBM | $0.99 \pm 0.01$ | $0.43 \pm 0.01$ | $0.01 \pm 0.01$ | $0.70 \pm 0.01$ | $0.01 \pm 0.01$ | $0.39 \pm 0.01$ | $0.91 \pm 0.01$ | $0.88 \pm 0.01$ |
| SL + DE | $0.99 \pm 0.01$ | $0.43 \pm 0.01$ | $0.01 \pm 0.01$ | $0.64 \pm 0.08$ | $0.01 \pm 0.01$ | $0.39 \pm 0.01$ | $0.87 \pm 0.05$ | $0.78 \pm 0.13$ |
| SL + bears | $0.99 \pm 0.01$ | $0.43 \pm 0.01$ | $0.01 \pm 0.01$ | $0.38 \pm 0.01$ | $0.01 \pm 0.01$ | $0.39 \pm 0.01$ | $0.75 \pm 0.01$ | $0.37 \pm 0.03$ |
| LTN | $0.98 \pm 0.01$ | $0.42 \pm 0.01$ | $0.02 \pm 0.01$ | $0.70 \pm 0.01$ | $0.06 \pm 0.01$ | $0.39 \pm 0.01$ | $0.94 \pm 0.01$ | $0.87 \pm 0.01$ |
| LTN + MCDO | $0.98 \pm 0.01$ | $0.42 \pm 0.01$ | $0.01 \pm 0.01$ | $0.69 \pm 0.01$ | $0.06 \pm 0.01$ | $0.39 \pm 0.01$ | $0.93 \pm 0.01$ | $0.87 \pm 0.01$ |
| LTN + LA | $0.98 \pm 0.01$ | $0.43 \pm 0.01$ | $0.14 \pm 0.02$ | $0.55 \pm 0.02$ | $0.06 \pm 0.01$ | $0.39 \pm 0.01$ | $0.79 \pm 0.02$ | $0.73 \pm 0.02$ |
| LTN + PCBM | $0.98 \pm 0.01$ | $0.43 \pm 0.01$ | $0.01 \pm 0.01$ | $0.69 \pm 0.01$ | $0.06 \pm 0.01$ | $0.39 \pm 0.01$ | $0.94 \pm 0.01$ | $0.86 \pm 0.01$ |
| LTN + DE | $0.99 \pm 0.01$ | $0.42 \pm 0.01$ | $0.01 \pm 0.01$ | $0.69 \pm 0.01$ | $0.06 \pm 0.11$ | $0.39 \pm 0.01$ | $0.94 \pm 0.01$ | $0.87 \pm 0.01$ |
| LTN + bears | $0.99 \pm 0.01$ | $0.43 \pm 0.01$ | $0.06 \pm 0.01$ | $0.36 \pm 0.01$ | $0.08 \pm 0.01$ | $0.39 \pm 0.01$ | $0.36 \pm 0.01$ | $0.32 \pm 0.01$ |

We include next the results on the MNIST-Even-Odd. Likewise, bears when paired to all NeSy models shows drastic improvements in terms of $ECE_Y$ and $ECE_C$, both *in* and *out-of-distribution*.

Table 11: Complete evaluation on MNIST-Even-Odd. All competitors struggle in terms of $Acc_Y$ *in-distribution* when not paired to SL, while bears shows sensible improvements when paired on both DPL and LTN. The accuracy on concepts $Acc_C$ *in-distribution* shows that all methods pick up a RS, despite being generally suboptimal. In the *out-of-distribution* we observe a drastic degradation on both $Acc_{Yood}$ and $Acc_{Cood}$.

| METHOD | $Acc_Y$ | $Acc_C$ | $ECE_Y$ | $ECE_C$ | $Acc_{Yood}$ | $Acc_{Cood}$ | $ECE_{Yood}$ | $ECE_{Cood}$ |
|---|---|---|---|---|---|---|---|---|
| | | | | MNIST-Even-Odd | | | | |
| DPL | $0.71 \pm 0.01$ | $0.01 \pm 0.01$ | $0.11 \pm 0.01$ | $0.81 \pm 0.01$ | $0.07 \pm 0.01$ | $0.07 \pm 0.01$ | $0.78 \pm 0.01$ | $0.85 \pm 0.01$ |
| DPL + MCDO | $0.72 \pm 0.01$ | $0.01 \pm 0.01$ | $0.09 \pm 0.01$ | $0.80 \pm 0.01$ | $0.07 \pm 0.01$ | $0.05 \pm 0.01$ | $0.77 \pm 0.01$ | $0.84 \pm 0.01$ |
| DPL + LA | $0.71 \pm 0.01$ | $0.01 \pm 0.01$ | $0.09 \pm 0.01$ | $0.78 \pm 0.01$ | $0.07 \pm 0.01$ | $0.01 \pm 0.01$ | $0.76 \pm 0.01$ | $0.83 \pm 0.01$ |
| DPL + PCBM | $0.78 \pm 0.08$ | $0.11 \pm 0.11$ | $0.15 \pm 0.13$ | $0.65 \pm 0.08$ | $0.05 \pm 0.03$ | $0.09 \pm 0.01$ | $0.71 \pm 0.15$ | $0.72 \pm 0.13$ |
| DPL + DE | $0.76 \pm 0.01$ | $0.01 \pm 0.01$ | $0.13 \pm 0.02$ | $0.69 \pm 0.06$ | $0.07 \pm 0.01$ | $0.05 \pm 0.01$ | $0.64 \pm 0.06$ | $0.70 \pm 0.07$ |
| DPL + bears | $0.93 \pm 0.03$ | $0.05 \pm 0.09$ | $0.21 \pm 0.03$ | $0.25 \pm 0.07$ | $0.03 \pm 0.03$ | $0.12 \pm 0.05$ | $0.46 \pm 0.03$ | $0.25 \pm 0.05$ |
| SL | $0.97 \pm 0.01$ | $0.01 \pm 0.01$ | $0.02 \pm 0.01$ | $0.82 \pm 0.01$ | $0.01 \pm 0.01$ | $0.07 \pm 0.01$ | $0.97 \pm 0.01$ | $0.87 \pm 0.01$ |
| SL + MCDO | $0.98 \pm 0.01$ | $0.01 \pm 0.01$ | $0.01 \pm 0.01$ | $0.80 \pm 0.01$ | $0.01 \pm 0.01$ | $0.05 \pm 0.01$ | $0.94 \pm 0.01$ | $0.85 \pm 0.01$ |
| SL + LA | $0.98 \pm 0.01$ | $0.01 \pm 0.01$ | $0.04 \pm 0.01$ | $0.73 \pm 0.01$ | $0.01 \pm 0.01$ | $0.01 \pm 0.01$ | $0.89 \pm 0.01$ | $0.78 \pm 0.01$ |
| SL + PCBM | $0.98 \pm 0.01$ | $0.01 \pm 0.01$ | $0.02 \pm 0.01$ | $0.83 \pm 0.01$ | $0.01 \pm 0.01$ | $0.07 \pm 0.01$ | $0.97 \pm 0.01$ | $0.88 \pm 0.01$ |
| SL + DE | $0.99 \pm 0.01$ | $0.01 \pm 0.01$ | $0.01 \pm 0.01$ | $0.77 \pm 0.07$ | $0.01 \pm 0.01$ | $0.05 \pm 0.01$ | $0.93 \pm 0.02$ | $0.81 \pm 0.08$ |
| SL + bears | $0.99 \pm 0.01$ | $0.01 \pm 0.01$ | $0.01 \pm 0.01$ | $0.34 \pm 0.02$ | $0.01 \pm 0.01$ | $0.07 \pm 0.02$ | $0.85 \pm 0.01$ | $0.33 \pm 0.03$ |
| LTN | $0.70 \pm 0.01$ | $0.28 \pm 0.05$ | $0.29 \pm 0.01$ | $0.64 \pm 0.05$ | $0.10 \pm 0.01$ | $0.09 \pm 0.01$ | $0.90 \pm 0.01$ | $0.79 \pm 0.01$ |
| LTN + MCDO | $0.72 \pm 0.01$ | $0.28 \pm 0.05$ | $0.21 \pm 0.02$ | $0.62 \pm 0.04$ | $0.11 \pm 0.01$ | $0.14 \pm 0.01$ | $0.85 \pm 0.01$ | $0.77 \pm 0.01$ |
| LTN + LA | $0.72 \pm 0.01$ | $0.24 \pm 0.14$ | $0.13 \pm 0.05$ | $0.61 \pm 0.12$ | $0.10 \pm 0.04$ | $0.26 \pm 0.09$ | $0.79 \pm 0.01$ | $0.74 \pm 0.03$ |
| LTN + PCBM | $0.73 \pm 0.01$ | $0.01 \pm 0.01$ | $0.27 \pm 0.01$ | $0.85 \pm 0.02$ | $0.01 \pm 0.01$ | $0.09 \pm 0.01$ | $0.99 \pm 0.01$ | $0.89 \pm 0.01$ |
| LTN + DE | $0.77 \pm 0.05$ | $0.23 \pm 0.07$ | $0.13 \pm 0.05$ | $0.32 \pm 0.05$ | $0.06 \pm 0.04$ | $0.08 \pm 0.04$ | $0.61 \pm 0.07$ | $0.43 \pm 0.10$ |
| LTN + bears | $0.89 \pm 0.06$ | $0.22 \pm 0.08$ | $0.11 \pm 0.02$ | $0.11 \pm 0.07$ | $0.13 \pm 0.02$ | $0.08 \pm 0.02$ | $0.22 \pm 0.03$ | $0.13 \pm 0.02$ |

## C.2 BDD-OIA

We report here the complete evaluation on BDD-OIA . The values on $\mathrm{mF_1}(Y)$ show that bears does not worsen sensibly the scores w.r.t. DPL and DPL paired with DE, despite being trained with an extra term (conflicting in principle with the optimization on label accuracy). LA and PCBM, on the other hand, perform worse compared to other methods. In terms of $\mathrm{mF_1}(C)$, both PCBM and bears improve the scores compared to DPL alone.

Table 12: Full results on BDD-OIA .

| | BDD-OIA | | | | | | |
|---|---|---|---|---|---|---|---|
| METHOD | $\mathrm{mF_1}(Y)$ | $\mathrm{mF_1}(C)$ | $\mathrm{mECE}_Y$ | $\mathrm{mECE}_C$ | $\mathrm{ECE}_C(F,S)$ | $\mathrm{ECE}_C(R)$ | $\mathrm{ECE}_C(L)$ |
| DPL | $0.72 \pm 0.01$ | $0.34 \pm 0.01$ | $0.08 \pm 0.01$ | $0.84 \pm 0.01$ | $0.75 \pm 0.17$ | $0.79 \pm 0.05$ | $0.59 \pm 0.32$ |
| DPL + MCDO | $0.72 \pm 0.01$ | $0.34 \pm 0.01$ | $0.07 \pm 0.01$ | $0.83 \pm 0.01$ | $0.72 \pm 0.19$ | $0.76 \pm 0.08$ | $0.55 \pm 0.33$ |
| DPL + LA | $0.67 \pm 0.03$ | $0.34 \pm 0.01$ | $0.12 \pm 0.03$ | $0.85 \pm 0.01$ | $0.84 \pm 0.10$ | $0.87 \pm 0.04$ | $0.67 \pm 0.19$ |
| DPL + PCBM | $0.68 \pm 0.01$ | $0.42 \pm 0.01$ | $0.12 \pm 0.01$ | $0.68 \pm 0.01$ | $0.26 \pm 0.01$ | $0.26 \pm 0.02$ | $0.11 \pm 0.02$ |
| DPL + DE | $0.72 \pm 0.01$ | $0.35 \pm 0.01$ | $0.10 \pm 0.01$ | $0.79 \pm 0.01$ | $0.62 \pm 0.03$ | $0.71 \pm 0.10$ | $0.37 \pm 0.12$ |
| DPL + bears | $0.70 \pm 0.01$ | $0.42 \pm 0.01$ | $0.06 \pm 0.01$ | $0.58 \pm 0.01$ | $0.14 \pm 0.01$ | $0.10 \pm 0.01$ | $0.02 \pm 0.01$ |

## C.3 KANDINSKY

We include here the evaluation curves for the active experiment on both the $\mathrm{Acc}_Y$ and $\mathrm{Acc}_C$ for DPL paired with the entropy strategy (in yellow), with the random baseline (in blue), and with bears (in red).

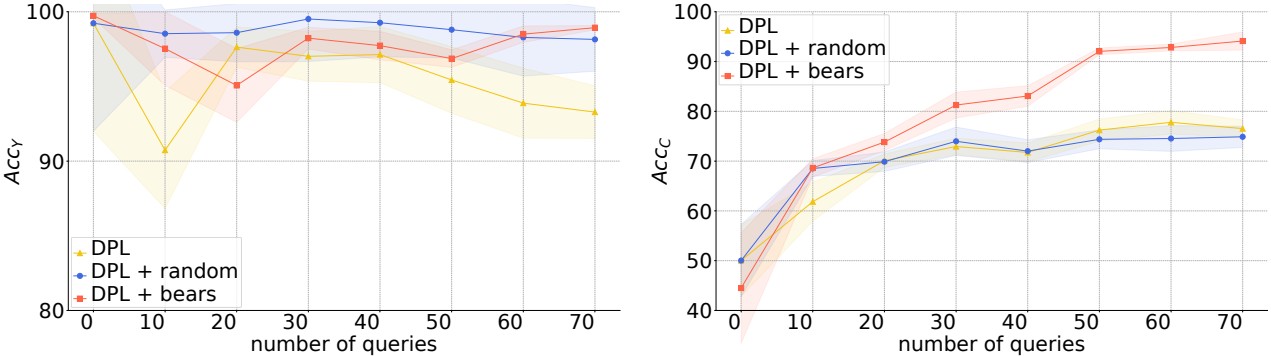

Figure 7: **bears allows selecting informative concept annotations faster.** (left) label accuracy. (*right*) concept accuracy.

## C.4 CONCEPT-WISE ENTROPY SCORES FOR `MNIST-HALF`

We report the entropy scores for each concept for all NeSy models we tested. `bears` performs as desired, whereas the runner-up, `LA`, struggles to put low-entropy on 0, especially when paired with `SL` and `LTN`.

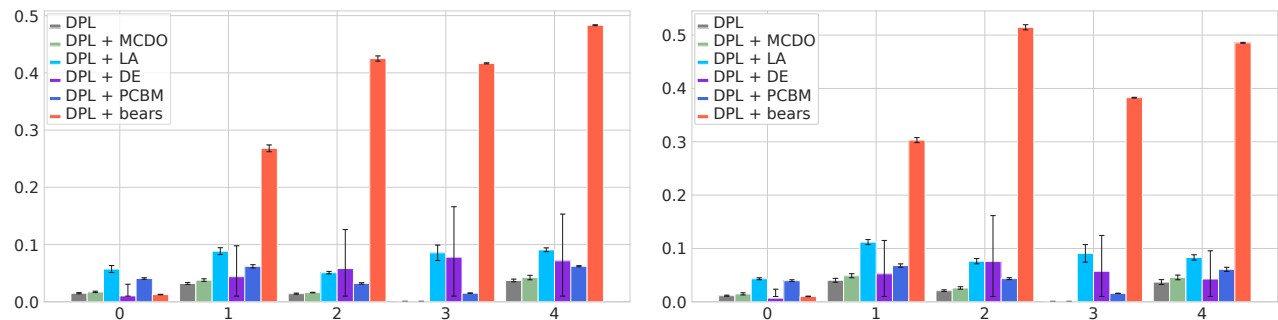

Figure 8: **DPL + bears shows high entropy for concepts affected by RSs while it does not for others in out-of-distribution settings.** (*left*) In distribution. (*right*) Out-of-distribution

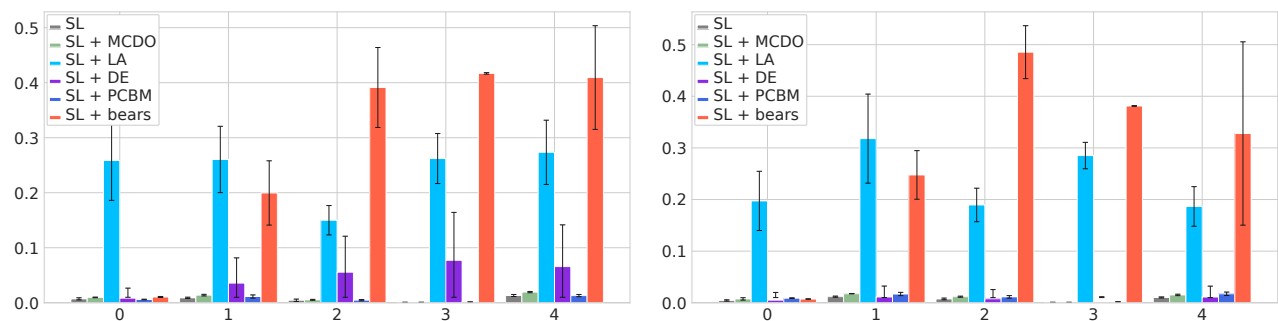

Figure 9: **SL + bears shows high entropy for concepts affected by RSs while it does not for others.** (*left*): in-distribution. (*right*): out-of-distribution.

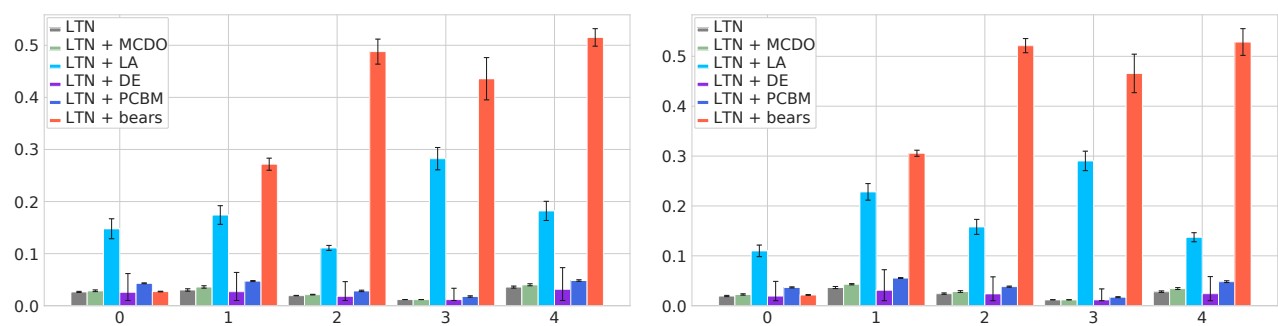

Figure 10: **LTN + bears shows high entropy for concepts affected by RSs while it does not for others.** (*left*): in-distribution. (*right*): out-of-distribution.

## C.5 CONFUSION MATRICES `KANDINSKY`

We report the confusion matrices (CMs) for the active learning experiment on `Kandinsky` dataset. At the beginning, `DPL` picks a RS showing that only few concept vectors **c** can be used to solve the classification task. At the last iteration, corresponding to collecting a total of 70 objects with concept annotation, `DPL` and `DPL + bears` show very different CMs. Both show that colors have learned correctly, although the concept annotation collected with `bears` make `DPL` align more to the diagonal, corresponding to the intended solution.

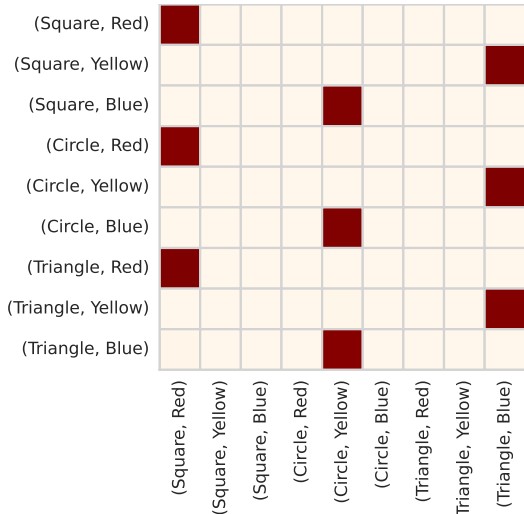

Figure 11: **`DPL` at iteration** 0 **in active learning settings on** `Kandinsky`

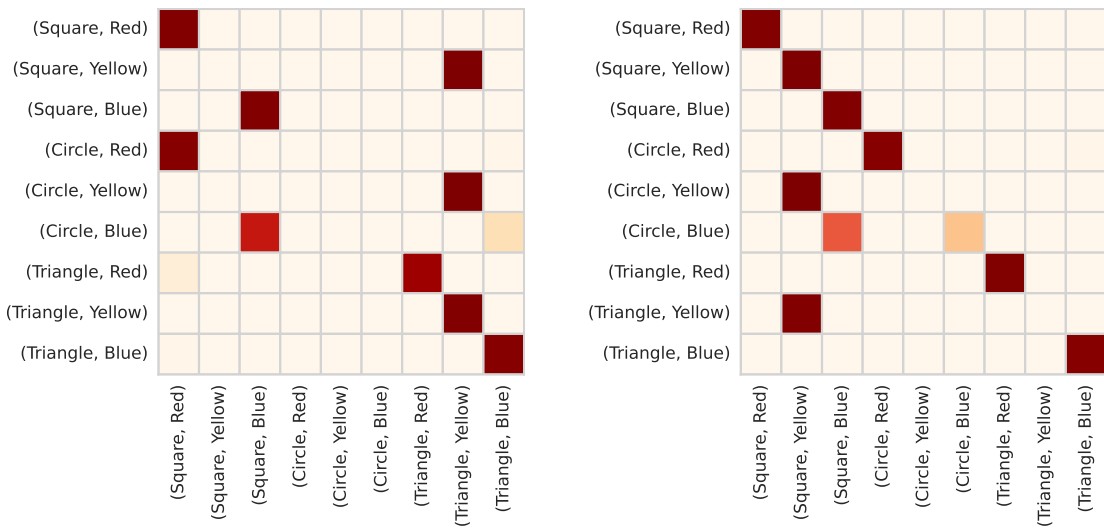

Figure 12: **Iteration** 70 **in active learning settings on** `Kandinsky`. Right: `DPL` Left: `DPL + bears`

## C.6 CONFUSION MATRICES ON BDD-OIA

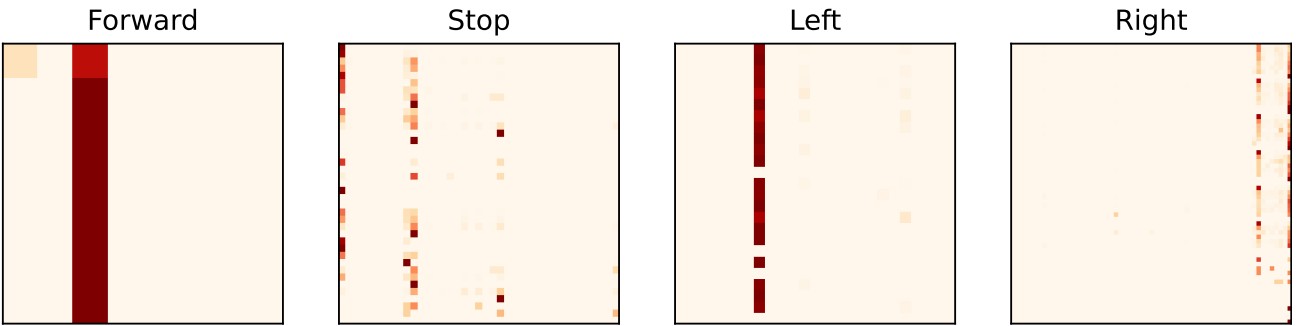

Figure 13: **DPL confusion matrices per concept classes on** BDD-OIA

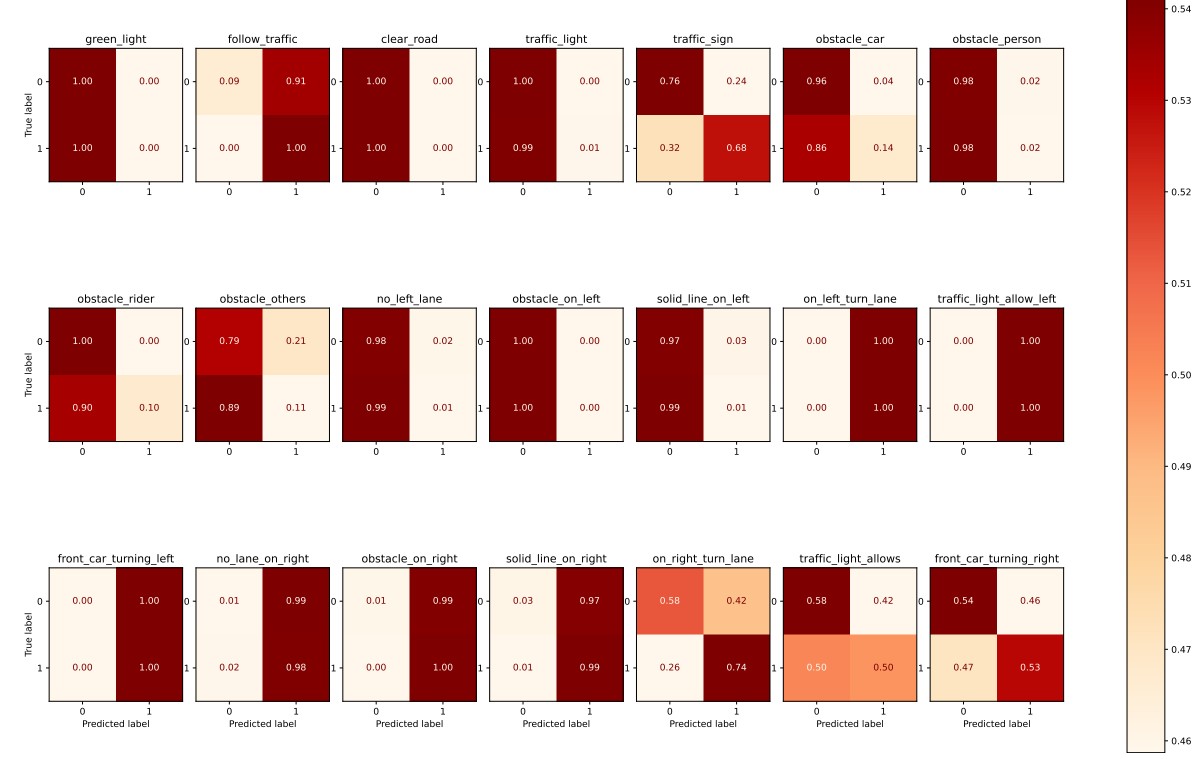

Figure 14: **DPL multilabel confusion matrix on** BDD-OIA

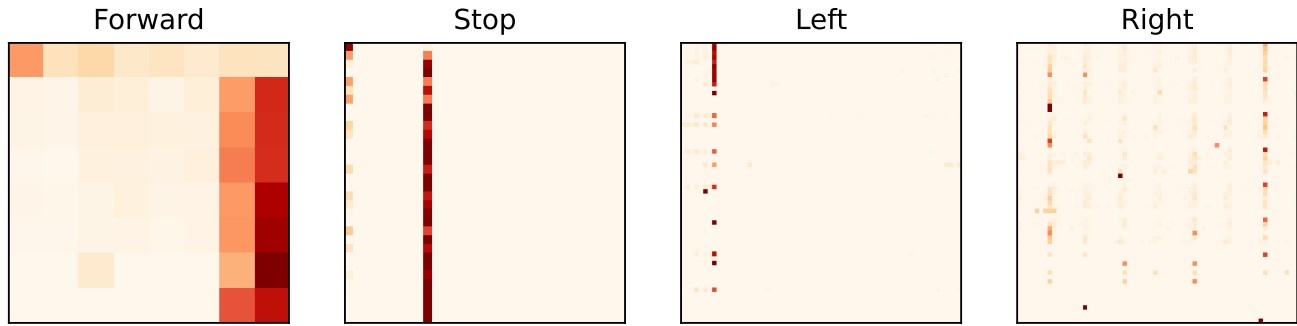

Figure 15: **DPL + bears** confusion matrices per concept classes on `BDD-OIA`

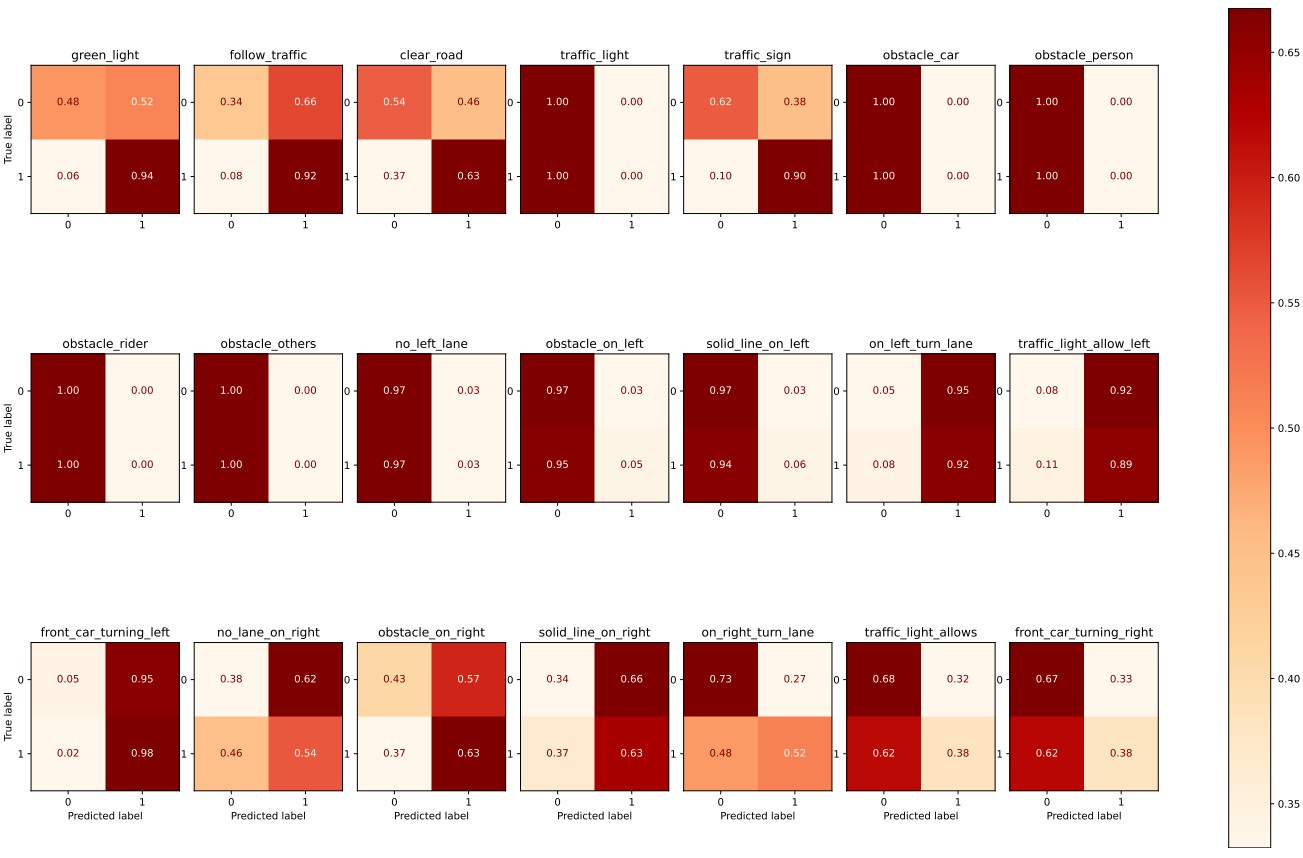

Figure 16: **DPL + bears** multilabel confusion matrix on `BDD-OIA`

## C.7  CONFUSION MATRICES ON `MNIST-HALF`

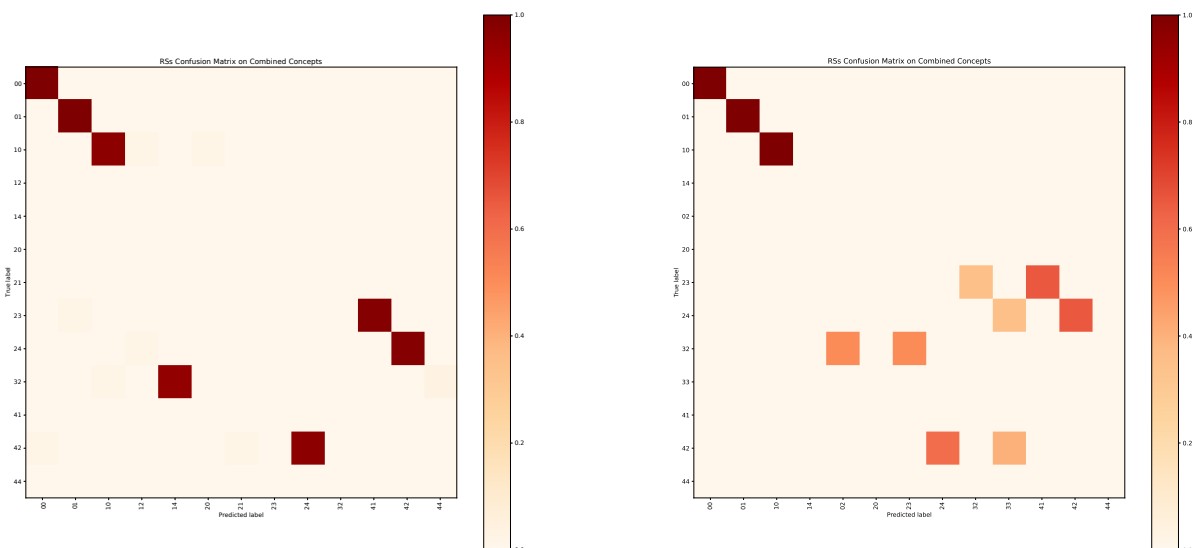

Figure 17: (*left*) **DPL** and (*right*) **DPL + `bears`** concepts confusion matrix on `MNIST-Half`

