# OpenReview forum: "BEARS Make Neuro-Symbolic Models Aware of their Reasoning Shortcuts"
_auai.org/UAI/2024/Conference — UAI 2024 spotlight_

### Official Review · Reviewer_VAQA · 2024-02-29

**Q2-1 Originality-Novelty:** 4
**Q2-2 Correctness-Technical Quality:** 3
**Q2-5 Clarity Of Writing:** 3

**Q1 Summary And Contributions:**

The paper presents a novel approach to reduce the effect of reasoning shortcuts in neuro-symbolic models. This is done by training an ensemble of models for concept extraction such that each model of the ensemble is affected by different shortcuts. In this way, shortcuts can be identified, increasing the performance of the models to which the system is applied.
Empirical results show that the proposed system significantly improves the performance of the model to which it is applied.

**Q2-3 Extent To Which Claims Are Supported By Evidence:**

3: Good: the main claims are supported by convincing evidence (in the form of adequate experimental evaluation, proofs, (pseudo-)code, references, assumptions).

**Q2-4 Reproducibility:**

4: Excellent: key resources (e.g. proofs, code, data) are available and key details (e.g. proof sketches, experimental setup) are comprehensively described for competent researchers to confidently and easily reproduce the main results.

**Q3 Main Strengths:**

The results are very good, information for reproducibility is available and the paper is clear and mathematically sound.

**Q4 Main Weakness:**

The paper was reduced by more than half in order to be submitted to the conference. Removing the details of implementation, of the model (which I still think are important though not essential in this kind of paper), and of the datasets, there are 15 pages of appendix. These pages contain important results that are mentioned in the main article and all the proofs. This seems a bit too many and risks reducing the real quality of the proposal. Without the appendix, the paper is almost incomplete.

**Q5 Detailed Comments To The Authors:**

I am very conflicted about the article. On the one hand it is clear and I found no particular errors. Furthermore, the results presented are very good. On the other hand, despite the work done by the authors to make the article enjoyable, it seems very incomplete to me. The appendix should serve to add less important details, not to add more than double the content to an article.

**Q9 Complying With Reviewing Instructions:**

Yes

---

> ### Author Rebuttal · Authors · 2024-04-07
>
> We thank the reviewer for going through our work in detail and for their positive comments, such as that the paper is clear and mathematically sound and the results are good. Below, we address the point raised in their review concerning the over-reliance on the appendix.
>
> **Experimental section lacks details.** Thank you for pointing out this issue. If accepted, we will use the two extra pages to move some material from the appendix to the main text to make it self-contained. In particular, we plan to add a paragraph on:
> * The metrics used, explaining their relation to the measures of our interests in their different versions (mean F1 and macro F1, expected calibration error ECE)
> * The implementation of DPL, SL, and LTN, as well as of a lengthier discussion on the Bayesian competitors
> * The design of the datasets, including further details on MNIST-Addition variations, an example of Kandinsky patterns from Figure 5, and the setup of BDD-OIA including the number of concepts and illustrative examples of the rules connecting them.
> * Improve the explanation of the results from the extended Tables 10 and 12. As we mentioned in the main text, all methods achieve similar F1 scores in all datasets, so we feel it is not necessary to include them explicitly in the main text.
>
> We hope this addresses the reviewer’s concern and ask for further feedback if more parts require the same treatment as above.

---

### Official Review · Reviewer_Sfww · 2024-03-21

**Q2-1 Originality-Novelty:** 3
**Q2-2 Correctness-Technical Quality:** 3
**Q2-5 Clarity Of Writing:** 4

**Q1 Summary And Contributions:**

When NeuroSymbolic models are trained without direct concept supervision, reasoning shortcuts can arise, meaning, learned concepts may diverge from their intended meaning. This work proposes an approach called bears to obtain models that are more aware of the potential concept ambiguity.


The proposed approach is to learn an ensemble of NeSy models. Importantly, the ensemble is learned by adapting the loss function to prefer a high likelihood **and** to disagree on the concept semantics with the other models in the ensemble. In this way, the ensemble is encouraged to cover the different possible interpretations. In other words, if reasoning shortcuts are possible, the models within the ensemble will cover the different concept interpretations. Concept ambiguity can then be detecting by considering the concept predictions of each model in the ensemble. As the ensemble is an average of the different models, the ambiguity for a concept will naturally lead to lower concept confidence.


Bears was empirically compared against others calibration techniques, on multiple NeSy systems. The evaluation shows very promising results: in the presence of reasoning shortcuts, Bears shows more awareness through a lower concept confidence, all while maintaining prediction accuracy.


As an extra experiment, they show how the now available concept ambiguity information can be used within active learning to focus on learning from instances, or concepts, that would reduce the ambiguity.

**Q2-3 Extent To Which Claims Are Supported By Evidence:**

3: Good: the main claims are supported by convincing evidence (in the form of adequate experimental evaluation, proofs, (pseudo-)code, references, assumptions).

**Q2-4 Reproducibility:**

4: Excellent: key resources (e.g. proofs, code, data) are available and key details (e.g. proof sketches, experimental setup) are comprehensively described for competent researchers to confidently and easily reproduce the main results.

**Q3 Main Strengths:**

Clarity of writing: very clearly written and motivated.

Simple, appealing idea with strong empirical results.

**Q4 Main Weakness:**

The used benchmarks are created to contain reasoning shortcuts, to more easily compare the different approaches. However, this raises the question of how much of a problem this is in reality?

**Q5 Detailed Comments To The Authors:**

To extend my question in Q4, it is for example not clear to me whether BDD-OIA was taken as is, or whether the task was altered/tested in a way that reasoning shortcuts are introduced. If it was taken as is, is it possible to provide an example of a reasoning shortcut that was found?

In Appendix, Figure 8, caption "(left" is missing a closing ). The legend in those plots is using "DPL + Ours" while previously "DPL + bears" was used.

**Q9 Complying With Reviewing Instructions:**

Yes

---

> ### Author Rebuttal · Authors · 2024-04-07
>
> We thank the reviewer for the positive comments about our work, particularly for finding it clear, bringing up simple appealing idea. Below, we address the reviewer question:
>
> **Benchmarks are crafted to contain reasoning shortcuts.** BDD-OIA was not altered in any way: it encompasses exactly the same knowledge and data provided in the original publication [1, 2]. It is possible to find an example of a reasoning shortcut learned by DPL in Figure 13 in Appendix C.6. From the figure, it happens that one forward concept (corresponding to “$\mathtt{follow-traffic}$”)  is always predicted as _TRUE_ while the stop concepts (corresponding to “$\mathtt{obstacle: car}$” and “$\mathtt{red-light}$”)  are used to choose between the forward and stop action. This highlights that the model does not use all concept combinations to correctly predict actions. A similar behavior is also evident in left and right confusion matrices.
>
> Variants of MNIST-Addition are the only cases where the dataset is filtered to ensure RSs are present.  Kandinsky patterns is also completely unfiltered.
>
> [1] Explainable Object-Induced Action Decision for Autonomous Vehicles, Xu et al., CVPR (2020) \
> [2] Not All Neuro-Symbolic Concepts Are Created Equal: Analysis and Mitigation of Reasoning Shortcuts, Marconato et al., NeurIPS (2023)
>
> **Typos.** Thank you for pointing us the typo, we will make sure to fix it.

---

### Official Review · Reviewer_Go7q · 2024-03-22

**Q2-1 Originality-Novelty:** 3
**Q2-2 Correctness-Technical Quality:** 4
**Q2-5 Clarity Of Writing:** 4

**Q1 Summary And Contributions:**

This submission proposes an alternative approach to handle the problem of reasoning shortcuts
that appear in neuro-symbolic applications where concepts of interest are not explicitly
mentioned in the training data, but are used only internally (hidden) for the reasoning aspect
of the model. Instead of solving the issue, the authors propose a method to detect it. Although
the implementation differs from the basic theory, the main intuition behind the method is to
maximise the entropy (that is, minimise the information) for the hidden concepts that suffer
from reasoning shortcuts. In a nutshell, this is made by combining the information of the
hidden concepts from models that showcase similar performance. If these hidden concepts are
different, then a reasoning shortcut took place. An experimental analysis shows that this
approach is effective.

**Q2-3 Extent To Which Claims Are Supported By Evidence:**

3: Good: the main claims are supported by convincing evidence (in the form of adequate experimental evaluation, proofs, (pseudo-)code, references, assumptions).

**Q2-4 Reproducibility:**

4: Excellent: key resources (e.g. proofs, code, data) are available and key details (e.g. proof sketches, experimental setup) are comprehensively described for competent researchers to confidently and easily reproduce the main results.

**Q3 Main Strengths:**

Novel view on the problem of reasoning shortcuts, with a sound approach that leverages ensembles of models to obtain a measure of certainty.

**Q4 Main Weakness:**

Does not fully solve the problems associated to reasoning shortcuts. It focuses on concepts, rather than their instances.

**Q5 Detailed Comments To The Authors:**

I found this work very interesting, and the approach, as far as I could verify is sound.
One interesting feature of the method is that, by combining different models into one, they
avoid the trap of trusting only one learned model, but take the information from many learners.
On the other side (but the authors are well aware of this) the approach does not solve the
issue with SR, and it remains open how to treat it. Using one of their examples, SR can be
dangerous if a rule says that one can pass a red light in an emergency. Well, knowing that
there was an SR still does not tell us whether we can pass the red light or not. This is
a problem because in case of emergency, we do not want to stop out of full uncertainty.

Another issue is that the approach assigns a low confidence to the *concept* not to its
*perception*; that is, red lights, and pedestrians are considered of low confidence in general,
but the issue could come from specific instances rather than with the concepts as a whole.
Mentions on this aspect would be useful.

Despite these comments, I believe that the work is important, and highly relevant to UAI and
as such I am suggesting acceptance.

**Q9 Complying With Reviewing Instructions:**

Yes

---

> ### Author Rebuttal · Authors · 2024-04-07
>
> We thank the reviewer for the positive remarks about our work, particularly for finding it interesting and highly relevant. Below we address the points raised by their review.
>
> **Awareness does not solve Reasoning Shortcuts.** We agree with the reviewer. Our starting point is how to model uncertainty on concepts with models that are affected by reasoning shortcuts, and that acquiring extra supervision on concepts is costly. With BEARS, it is possible to be aware of the concepts that are impacted by reasoning shortcuts, which is valuable information that can be used to decrease the amount of concept supervision. Our active learning experiment goes in this direction.
>
> **The difference between concept confidence and instance confidence.** Thank you for bringing up this point. We agree that some instances **x** may be misclassified and predicted with high-confidence, without strongly affecting the confidence on concepts. If that is the case, however, the expected calibration error would increase, as it penalizes instances **x** that are wrongly predicted with high confidence. We will mention this point in the discussion on the evaluation metrics.

---

### Official Review · Reviewer_3fQX · 2024-03-23

**Q2-1 Originality-Novelty:** 3
**Q2-2 Correctness-Technical Quality:** 3
**Q2-5 Clarity Of Writing:** 3

**Q1 Summary And Contributions:**

The paper introduces bears (BE Aware of Reasoning Shortcuts), a method designed to make Neuro-Symbolic (NeSy) models aware of reasoning shortcuts (RSs), which are instances where models achieve high accuracy by learning unintended semantics. This issue undermines the reliability and generalization of NeSy models. The authors address this by calibrating the model's concept-level confidence, thereby encouraging uncertainty about concepts affected by RSs, without compromising prediction accuracy. The approach leverages a diversified ensemble specifically trained to adjust the concepts' uncertainty in proportion to their susceptibility to RSs.

**Q2-3 Extent To Which Claims Are Supported By Evidence:**

4: Excellent: all claims are supported by very convincing evidence (in the form of comprehensive experimental evaluation, rigorous mathematical proofs, detailed (pseudo-)code, precise references, well-motivated and realistic assumptions) and the authors deliver what they promise.

**Q2-4 Reproducibility:**

4: Excellent: key resources (e.g. proofs, code, data) are available and key details (e.g. proof sketches, experimental setup) are comprehensively described for competent researchers to confidently and easily reproduce the main results.

**Q3 Main Strengths:**

- The paper identifies a novel phenomenon: the magnification of overconfidence issues in neuro-symbolic models. It underscores the importance of addressing this by calibrating models to foster uncertainty towards equivalent concepts, which is a crucial insight.
- The derivation and rationale behind the training objective presented in Equation 7 are thoughtfully developed, reflecting a strong conceptual foundation for the proposed approach.
- This paper draws an insightful connection between the identified phenomenon and active learning strategies, highlighting how the proposed method can effectively diminish the costs associated with acquiring annotations for concept supervision.
- The experiments conducted are comprehensive, validating the efficacy of the proposed approach in addressing the identified issue.

**Q4 Main Weakness:**

- While the theoretical analysis provides valuable insights into the underpinnings of the proposed training objective, it appears to be offering theoretical motivation rather than a performance guarantee. Could you discuss under what conditions the proposed method will succeed or fail?

**Q5 Detailed Comments To The Authors:**

- Regarding the data generating process, the assumption of invertibility posits a direct mapping from ground-truth concepts $G$ to observable data $X$, signified by the arrow $G \rightarrow X$. This raises an interesting question about the potential reversibility of this relationship, specifically whether the scenario $G \leftarrow X$ could be reasonable. Your insights on this matter would be greatly appreciated.
- The exploration of reasoning shortcuts and likelihood appears to align with the rank criterion for abductive knowledge induction as discussed in [1] [2]. While the rank criterion is designed for multi-instance settings, I am wondering whether it can apply to the single-instance setting considered in this paper, especially considering the similar challenges they present. Could you share some thoughts?



[1] Tao, L., Huang, Y. X., Dai, W. Z., & Jiang, Y. (2023). Deciphering Raw Data in Neuro-Symbolic Learning with Provable Guarantees. arXiv preprint arXiv:2308.10487.
[2] Dai, W. Z., & Muggleton, S. H. (2020). Abductive knowledge induction from raw data. arXiv preprint arXiv:2010.03514.

**Q9 Complying With Reviewing Instructions:**

Yes

---

> ### Author Rebuttal · Authors · 2024-04-07
>
> We thank the reviewer for the positive remarks about our work, particularly for finding it novel and conceptually founded, and appreciating the link with active learning. Below, we address their questions.
>
> **Performance guarantees of BEARS.** The reviewer is correct, our analysis does not provide statistical guarantees about the uncertainty of concepts and final task performance, although we believe it provides a venue for interesting future research. Our primary objective was to reconcile uncertainty awareness and reasoning shortcuts and show it can be done in a principled manner by averaging over different models with BEARS. Statistical learning bounds for the neural component in NeSy have recently been studied in [1] and in the paper the reviewer suggested to us [2], although it is not immediately clear how these can be related to RSs and uncertainty estimation. We will mention this work in the conclusion as important future work.
>
> [1]  Wang et al., On Learning Latent Models with Multi-Instance Weak Supervision, NeurIPS (2023) \
> [2] Tao, L., Huang, Y. X., Dai, W. Z., & Jiang, Y. (2023). Deciphering Raw Data in Neuro-Symbolic Learning with Provable Guarantees. arXiv.
>
> **The data generation process.** We thank the reviewer for raising this interesting point. The data generation process with the arrow $G \to X$ is commonly used in causal representation learning. In our case, by leveraging assumption **A1**, the relation between G and X is given by $G=f(X)$. Since we only deal with “observational” data, it is not possible to distinguish whether $G \to X$ holds or $X \to G$, and one can equivalently write $p(x,g)=p(g | x) p(x) = p(x | g) p(g)$, so both arrows give the same observed distribution. Conversely, if one had “interventional” data, the direction of the arrow would matter. Using interventional data is studied in Causal Representation Learning, e.g. [3, 4], and is also an interesting future research venue for understanding Reasoning Shortcuts.
>
> [3]  Towards Causal Representation Learning, Schoelkopf et al., IEEE (2021) \
> [4] Nonparametric Identifiability of Causal Representations from Unknown Interventions, von Kugelgen, NeurIPS (2023)
>
> **Connection to Abductive Learning.** We thank the reviewer for pointing us to these works, and we make sure to cite them in the related work. We are not familiar with this line of work, although abductive learning seems helpful to _mitigate_ reasoning shortcuts. BEARs is agnostic of the specific NeSy model that is used on top and it would support naturally also abductive methods. Combining BEARs and abductive learning has the potential to improve further uncertainty quantification, by leveraging natural mitigation offered by the abduction learning part. We see this as a promising future work direction.

---

### Official Review · Reviewer_Me8G · 2024-03-27

**Q2-1 Originality-Novelty:** 3
**Q2-2 Correctness-Technical Quality:** 3
**Q2-5 Clarity Of Writing:** 3

**Q1 Summary And Contributions:**

The paper proposes to consider cofounding factors when designing a NeSy system by isolating the predictors and the reasoner in the predictions. This is an interesting work, that uncovers a major problem of NeSy systems and that was not really addressed since now. The approach is nicely illustrated with some classical examples (code provided). The level of reasoning proposed in the approach is very shallow and the question of scaling up the approach is remaining.

**Q2-3 Extent To Which Claims Are Supported By Evidence:**

3: Good: the main claims are supported by convincing evidence (in the form of adequate experimental evaluation, proofs, (pseudo-)code, references, assumptions).

**Q2-4 Reproducibility:**

4: Excellent: key resources (e.g. proofs, code, data) are available and key details (e.g. proof sketches, experimental setup) are comprehensively described for competent researchers to confidently and easily reproduce the main results.

**Q3 Main Strengths:**

The paper is well written and well illustrated, even if relying too much on the supplementary materials for the details. The problem addressed is important and not directly addressed in the classical NeSy approaches. Questions are clears and answers convincing in general.
Even if the topic is not yet solved, this is a very interesting path to open.

**Q4 Main Weakness:**

The experimentation part is impossible to follow without the supplementary materials. I would rewrite this part to make it self-content. In addition to the ECEy value I would simply also report the F1-values for all the labels.
The paper indeed illustrates the work with the potential confusion of red-light and pedestrians in a simple example. Then they use a classical benchmarks for self-driving decision systems. It would be helpful to read whether this hypothesis holds or not for instance on the BDD-OIA.

The other weak point is the shallow nature of the reasoning part of the approaches. Final predictions (concepts) are very simple combinations of labels. It would be interesting to question the scaling of the approach (in particular the question of convex compositions) when the reasoning part is more complex.

**Q5 Detailed Comments To The Authors:**

As reported above, the topic is very interesting and the composition of probabilities for final classification may have a cofounding part that may hurt the predictions.

The paper is well written but a little bit dense and some of the materials that are needed are in the supplementary materials (it is impossible to follow the experimental part without the appendix).

The term “RS-Awareness” is may be a little bit too strong, the system is only trained in particular way.

I am also questioning, in the weak part of the approach, the scaling capabilities of the solution. Bears grows an ensemble of distinct concept distributions. How well does that scales for more complex tasks?

**Q9 Complying With Reviewing Instructions:**

Yes

---

> ### Author Rebuttal · Authors · 2024-04-07
>
> We thank the reviewer for their interest in our work and for finding it important and strong. Below we address the points they raised.
>
> **The experimental section lacks detail.** Thank you for pointing out this issue. If accepted, we will use the two extra pages to move some material from the appendix to the main text to make it self-contained. In particular, we plan to add a paragraph on:
> * The metrics used, explaining their relation to the measures of our interests in their different versions (mean F1 and macro F1, expected calibration error ECE)
> * The implementation of DPL, SL, and LTN, as well as of a lengthier discussion on the Bayesian competitors
> * The design of the datasets, including further details on MNIST-Addition variations, an example of Kandinsky patterns from Figure 5, and the setup of BDD-OIA including the number of concepts and illustrative examples of the rules connecting them.
> * Improve the explanation of the results from the extended Tables 10 and 12. As we mentioned in the main text, all methods achieve similar F1 scores in all datasets, so we feel it is not necessary to include them explicitly in the main text.
>
> **Shallow nature of reasoning in the experiments.**  We agree it would be very interesting to evaluate the models on more challenging datasets, however we made use of NeSy benchmarks that were used to test the current SotA for RS [1]. Despite their apparent simplicity, our tasks are not trivial from a reasoning perspective (MNIST-Addition involves hundreds of clauses in CNF; BDD-OIA constraints across concepts; and Kandinsky patterns multiple reasoning steps, i.e. first aggregating objects in the image and then aggregating different images) and this is sufficient for RSs to emerge.
>
> [1] Not All Neuro-Symbolic Concepts Are Created Equal: Analysis and Mitigation of Reasoning Shortcuts, Marconato et al., NeurIPS (2023)
>
> **Scaling BEARS to complex tasks.** BEARS is NeSy predictor-agnostic, thus scaling depends on both (1) efficiency of inference of the underlying NeSy predictor, and (2) size of the ensemble needed for proper uncertainty calibration.
>
> 1 is outside of our control. As for 2, BEARS requires only _one_ call to the reasoning engine, as we can always average the concept probabilities over the ensemble _before_ passing them to the reasoning layer.
>
> In future work, we plan to further scale BEARS by relying on embeddings-based reasoning layers [2] and by replacing the ensemble with a set of multiple heads sitting on top of the same neural backbone.
>
> [2] Soft-Unification in Deep Probabilistic Logic, Maene and De Raedt, NeurIPS (2023)
>
> **The running example and relation to BDD-OIA.** Our example is indeed a simplification of what happens in BDD-OIA.  For instance, consider the RSs affecting DPL shown in Figure 13, Appendix C.6: the  "$\mathtt{follow-traffic}$" concept is always predicted as TRUE, while “$\mathtt{obstacle: car}$” and “$\mathtt{red-light}$” are used to switch between predicting the forward and stop actions. This highlights that the model does not use all concept combinations to correctly predict actions (see the confusion matrix in Fig 13). Similar behavior is also evident from the “$\mathtt{turn\  left}$” and “$\mathtt{turn\ right}$” concept confusion matrices.
>
> **RS-Awareness may be too strong.** ​​We are not sure what the reviewer means exactly.  Our definition of RS-awareness can be found in Section 3, divided into three desiderata. BEARs is a practical implementation - with a solid theoretical motivation - that encourages RS-awareness by minimizing a loss term that increases the entropy of those concepts affected by RSs, hence lowering confidence and meeting D1.

---

### Meta-Review · Area_Chair_JgRU · 2024-04-20

Thank you for your submission.  Reviewers agreed that this paper makes a significant advance in addressing challenges due to reasoning shortcuts and concept ambiguity in neurosymbolic models.  Reviewers found the approach novel and sound and found the experimental results compelling.  Reviewers recommended reworking the balance of material between the appendix and the main paper; and authors in rebuttal agreed.